# Peak learning of mass spectrometry imaging data using artificial neural networks

Walid M. Abdelmoula [1], Begona Gimenez-Cassina Lopez[1], Elizabeth C. Randall[2], Tina Kapur[2], Jann N. Sarkaria [3], Forest M. White [4], Jeffrey N. Agar [5], William M. Wells[2,6] & Nathalie Y. R. Agar [1,2,7] ✉

Mass spectrometry imaging (MSI) is an emerging technology that holds potential for improving, biomarker discovery, metabolomics research, pharmaceutical applications and clinical diagnosis. Despite many solutions being developed, the large data size and high dimensional nature of MSI, especially 3D datasets, still pose computational and memory complexities that hinder accurate identification of biologically relevant molecular patterns. Moreover, the subjectivity in the selection of parameters for conventional pre-processing approaches can lead to bias. Therefore, we assess if a probabilistic generative model based on a fully connected variational autoencoder can be used for unsupervised analysis and peak learning of MSI data to uncover hidden structures. The resulting msiPL method learns and visualizes the underlying non-linear spectral manifold, revealing biologically relevant clusters of tissue anatomy in a mouse kidney and tumor heterogeneity in human prostatectomy tissue, colorectal carcinoma, and glioblastoma mouse model, with identification of underlying m/z peaks. The method is applied for the analysis of MSI datasets ranging from 3.3 to 78.9 GB, without prior pre-processing and peak picking, and acquired using different mass spectrometers at different centers.

[1] Surgical Molecular Imaging Laboratory, Department of Neurosurgery, Brigham and Women's Hospital, Harvard Medical School, Boston, MA 02115, USA. [2] Department of Radiology, Brigham and Women's Hospital, Harvard Medical School, Boston, MA 02115, USA. [3] Department of Radiation Oncology, Mayo Clinic, 200 First St SW, Rochester, MN 55902, USA. [4] Department of Biological Engineering, Koch Institute for Integrative Cancer Research, MIT, Cambridge, MA 02142, USA. [5] Department of Chemistry and Chemical Biology, Northeastern University, 412 TF (140 The Fenway), Boston, MA 02111, USA. [6] Computer Science and Artificial Intelligence Laboratory, MIT, Cambridge, MA 02139, USA. [7] Department of Cancer Biology, Dana-Farber Cancer Institute, Harvard Medical School, Boston, MA 02115, USA. ✉email: Nathalie_Agar@dfci.harvard.edu

Mass spectrometry imaging (MSI) is a rapidly growing technology that holds high promise to impact the practice of anatomic pathology and drug development[1–3]. MSI provides simultaneous mapping of hundreds to thousands of molecules directly from a tissue section in a label free manner[4]. Moreover, MSI can provide direct molecular imaging of multiple types of molecules, such as proteins, peptides, lipids, metabolites, and drug molecules, with high sensitivity and molecular specificity. These molecular data can play a substantial role for improving clinical diagnosis and prognosis[5], pathway identification[6], biomarker discovery[7], and surgical guidance[3]. Sample preparation, ionization techniques, and instrumentation are determinants for the effectiveness of analyte detection. Matrix-assisted laser desorption ionization (MALDI) and derivatives of electrospray ionization (ESI) such as desorption (DESI) and continuous flow surface sampling are among the most common ionization techniques for MS towards clinical applications[1]. For a mass spectrometry image acquisition, molecules are desorbed and ionized from the surface of a sample, and then separated in a mass analyzer based on their mass-to-charge ratio ($m/z$) and detected to measure their relative abundance forming a mass spectrum.

MSI data are large and complex. For instance, a raw high mass resolution MSI file size can reach up to a few terabytes of spectral information. Computational developments that would more efficiently and accurately mine MSI data to identify molecular signatures of clinical importance and enable new biomarker discovery have the potential to expand the applicability of MSI[8,9]. However, the complex nature of MSI data hinders efficient data mining, clustering, visualization, and classification using traditional machine learning techniques[10,11]. This data complexity poses memory and computational challenges, namely due to "the curse of dimensionality" in which original MSI data hold up to tens of thousands of spectra each of which has $10^4-10^6$ $m/z$ spectral bins, and the nonlinear separability of the underlying spectral manifold in the high-dimensional space.

Peak picking is currently a fundamental data preprocessing step for the analysis of original mass spectral data at the basis of biomarker discovery[12,13]. Peak picking is used to alleviate the sparsity and reduce the original spectral dimensionality while optimally increasing the signal-to-noise ratio through retaining as many informative $m/z$ features as possible. In addition, peak picking is essential for identification, quantification, and discovery of molecular biomarkers[14]. Despite the generally acceptable performance of peak picking algorithms, each applied parameter, e.g., baseline subtraction, peak width, signal-to-noise ratio (S/N), and smoothing introduces a level of subjectivity that influences the resulting peak list[15]. The optimization of parameter selection largely relies on the user's expertise and can therefore lead to significant discrepancy in overall biomarker identification[16]. These limitations are exacerbated when applying these workflows to MSI data, which is large and has an added level of complexity with spatial information[17].

Following peak picking the original dimensional complexity is reduced; however, MSI data are still of high-dimensional nature as one 2D image is typically composed of thousands of high-dimensional pixels (spectra) each of which has hundreds of peaks. High-dimensional statistics for dimensionality reduction are commonly used[18,19]. Dimensionality reduction aims at projecting the high-dimensional points into a smaller subspace to enable the capture and visualization of the underlying latent variables. Those latent variables reveal molecular patterns, reflecting clusters of similar spectra that might hold biological relevance[20]. Linear dimensionality reduction methods of principal component analysis (PCA) and non-negative matrix factorization (NNMF) have been widely used for MSI data analysis[21,22]. A limitation of these methods is their inherited linearity constraints (e.g., the original data are linearly mapped based on a linear combination of lower-dimensional vectors) that prevent capturing the complex nonlinear manifold of spectral structures, impacting accurate identification of latent variables. In contrast, nonlinear dimensionality reduction methods such as t-distributed stochastic neighbor embedding (t-SNE) have gained popularity in the last few years for omics data analysis[20,23,24]. Nevertheless, t-SNE does not provide parametric mapping needed to project new unseen data into the already computed embedding. Despite recent progress on improving both the t-SNE computational and memory scalability[25,26], it still needs the full data to be loaded into the RAM, which limits its application on data with large sizes such as 3D MSI[27]. Loading the full data are necessary for the K-nearest neighbor graph creation, which is instrumental in establishing spectral pairwise similarities to compute the final t-SNE embedding. Recent preliminary results, on a single preprocessed 2D MSI dataset that underwent peak picking, produced by using a neural-network-based method of autoencoder have shown promise for efficient nonlinear dimensionality reduction of MSI data compared to PCA and NNMF methods[28]. We propose to extend this neural-network approach by developing a deep learning architecture that can analyze MSI data, without prior peak picking, independent of the nature of the specimen and of the mass spectrometer (ionization source and analyzer).

We introduce msiPL, a deep learning tool for the analysis and peak learning of MSI data, which is based on a fully connected variational autoencoder neural network[29]. This is a probabilistic generative model that learns unsupervised and nonlinear parametric mapping between high and low-dimensional spaces and has been efferently applied to other fields such as single cell omics[30] and medical image segmentation[31]. The low-dimensional embedding learns a nonlinear manifold that captures latent variables that we refer to as encoded features. The encoded features represent molecular patterns that are used to predict the original data. Therefore, minimizing the error between original and predicted data would imply capturing accurate encoded features. Batch normalization is incorporated into the proposed neural-network architecture to correct for co-variate shift and improve both learning stability and convergence[32]. We also propose a method based on analyzing the neural-network weight matrix to relate the encoded features to the original $m/z$ features. Both the encoded features and their associated $m/z$ ion features would support clustering and classification tasks required for biomarker identification. The performance of the proposed method was tested using various 2D and 3D MSI data of biological samples collected from different organs and acquired in different laboratories using different MSI platforms. Namely, MSI data from both human and animal tissue specimens were acquired using: MALDI TOF MSI, DESI MSI, and 9.4 Tesla MALDI FT-ICR MSI with some of these datasets already publicly available[27] and some newly acquired datasets (see Project ID (2703) on the metabolomics workbench https://www.metabolomicsworkbench.org).

## Results

**Variation inference for manifold learning of MSI data.** The computational architecture, shown in Fig. 1, is based on a probabilistic generative model to establish efficient unsupervised learning, nonlinear dimensionality reduction and stochastic variational inference using neural networks[29]. The variational autoencoder (VAE) aims to jointly optimize two distinct models, namely: probabilistic encoder for variational inference and probabilistic decoder for unsupervised learning. The probabilistic encoder is used as a recognition model $q_\phi(z|x)$ to infer approximate estimate of the true but intractable distribution $p_\Theta(z|x)$ of the $k$-dimensional latent variable $z$ (i.e., encoded features) underlying the complex high-dimensional MSI data

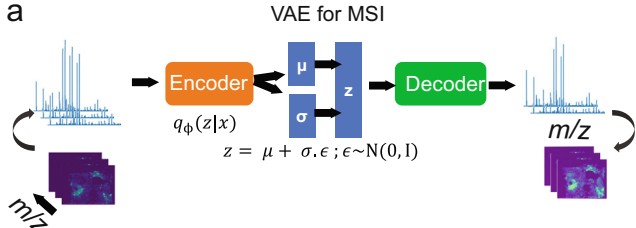

**a** VAE for MSI

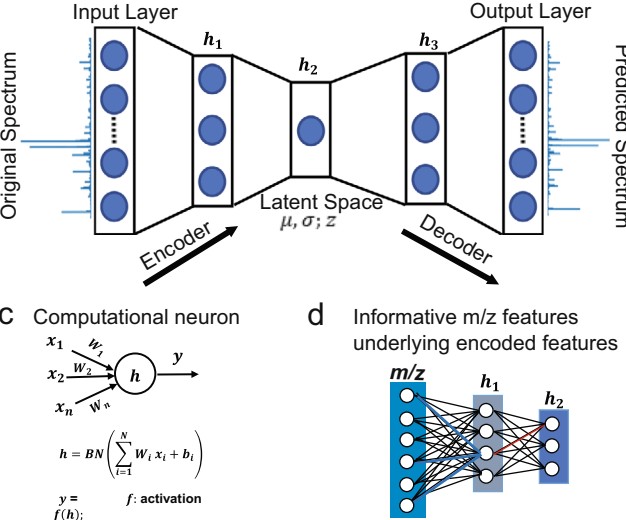

**b** Fully connected neural network of VAE with 3 hidden layers:

**c** Computational neuron

$$h = BN\left(\sum_{i=1}^{N} W_i x_i + b_i\right)$$

$$y = f(h);$$

$f$: activation

**d** Informative m/z features underlying encoded features

**Fig. 1 The proposed neural-network architecture of variational autoencoder (VAE) for mass spectrometry imaging data analysis and peak learning. a** Schematic overview of the VAE model, which was imeplemnted as a fully connected neural network (**b**) of five layers and trains on TIC-normalized spectra without considering their spatial relationships (**b**). The parametrized lower-dimensional latent variable (Z) is captured at hidden layer $h_2$. **c** The neural network is regularized using batch normalization (BN), and informative mass-to-charge ratio (m/z) values were identified using statistical analysis on the neural-network weight parameter (**d**).

$X = \{x^{(1)}, x^{(2)}, \ldots, x^{(N)}\}$, where $N$ represents the total number of spectra and each spectrum $x^{(i)} \epsilon \mathbb{R}^d$ is of $d-$dimensions, and $d \gg k$. The probabilistic encoder model assumes sampling of the latent variable $z$ from a multivariate normal distribution with parameters $\mu_\phi(x)$ and $\sigma_\phi(x)$; both are functions of $x$ and computed by a fully connected neural network. The probabilistic decoder is used as a generative model $p_\Theta(x|z)$ for data reconstruction given solely the encoded features. Both the recognition and generative models' parameters, $\phi$ and $\Theta$, are jointly optimized and computed from the neural-network parameters. More details are provided in the Methods section and we refer to ref. [29] for more information on VAE.

**Hyperparameters and implementation details**. Original MSI data were analyzed using msiPL, a VAE architecture of a fully connected neural network given in Fig. 1b. The proposed neural-network architecture consists of five layers, namely: input layer $(L_1)$, three hidden layers $(h_1, h_2, \text{and } h_3)$, and output layer $(L_5)$. The number of artificial neurons for both $L_1$ and $L_2$ is equivalent to the number of m/z bins, whereas for the hidden layers $h_1, h_2, \text{and } h_3$ it is 512, 5, 512 neurons, respectively. The hidden layer $h_2$ captures the encoded features, which represent nonlinear dimensionality reduction of original MSI data and compressed in

a five-dimensional space. Batch normalization was applied on each layer's input before any neuron activation to correct for co-variate shift that would degrade the learning process[32]. Following the batch normalization, the rectified linear unit (ReLU) function was used for neuron activation in all layers except the neurons of the output layer $L_5$, which were activated using the sigmoid function[33]. The unsupervised learning occurs through minimizing the reconstruction loss between original and predicted data mainly by optimizing the VAE cost function, which consists of Kullback-Leibler divergence (KL-divergence) and marginal likelihood modeled as a categorical cross-entropy[29]. The Adam stochastic gradient optimizer with learning rate of 0.001 was used to train the network on minibatches of 128 spectra for 100 epochs[34]. This network was trained on *total-ion-count* (*TIC*) normalized spectra and implemented using the open source deep learning library of Keras[35] and running on Tensorflow[36].

**Linking encoded features to observed m/z variables**. The encoded features represent the learned nonlinear manifold in the lower-dimensional space, and enabled capturing spatial patterns of molecules from the original high-dimensional space. These patterns were formed based on a smaller subset of m/z features and it is therefore of interest to identify those underlying m/z features that are expected to hold biological relevance. We propose a backpropagated-based threshold analysis on the weight parameter $W^{(L)}$ of the neural network at layer $L$, as visually illustrated in Fig. 1d and demonstrated in Eq. (4). Since the original MSI data were analyzed without prior preprocessing for peak picking, the identified m/z features can then be assigned to a peak. As such, a peak is identified on the average spectrum as the nearest local maximum of a given m/z feature. More details are given in the Methods section.

**Analysis of FT-ICR MSI data from human prostate cancer tissue specimen**. Ultrahigh spectral resolution 2D FT-ICR MSI data from a human prostate cancer specimen was computationally analyzed using msiPL. The original MSI data encompassed 12,716 pixels (spectra) and each pixel is a high-dimensional datapoint that contained 730,403 m/z values (dimensions) for the mass range m/z 250–1000. The data were exported in the standardized format imzML[37] and converted to HDF5 format[38] (using the python package "h5py") for variational autoencoder analysis. The original dataset was highly sparse and with enormous dimensionality and comprised 730,403 m/z values causing computational and memory challenges to optimize the 748 million parameters of the proposed neural network. To reduce the sparsity and the large number of m/z values and avoid allocating unnecessary memory, local maxima were identified on the average spectrum using the python function "argrelextrema" in which a local maximum is defined as a datapoint of intensity higher than its two neighbors. This significantly reduces the spectral dimensionality from approximately 730,403 to 61,343 m/z values, and this has significantly impacted the spectral sparsity but not the spectral representation (See Supplementary Fig. 1). The neural network performs unsupervised learning in an iterative manner to minimize the reconstruction loss and as shown in Fig. 2a, the optimizer converged after less than 100 iterations with a total running time of about 40 min on a PC workstation (Intel Xenon 3.3 GHz, 512 GB RAM, 64-bit Windows, 2 GPUs NVIDIA TITAN Xp). The distributions of *TIC*-normalized average spectra of both original and predicted data are given in Fig. 2b and their overlay reflects high estimation quality. The encoded features, shown in Fig. 2c, serve as a nonlinear embedding that enables visualization and reveals molecular patterns embedded in a compact representation of the original high-dimensional data

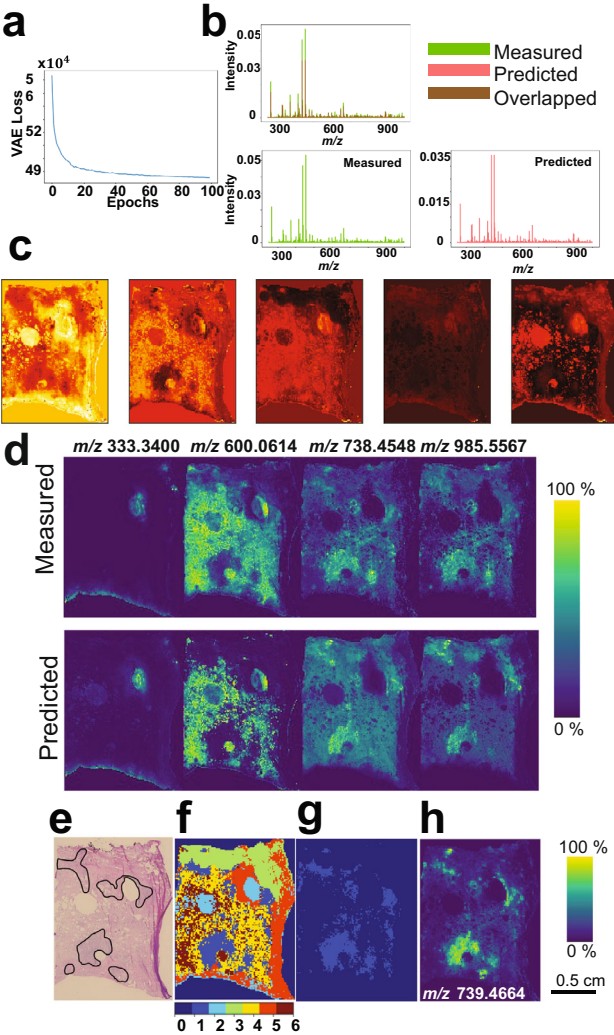

**Fig. 2 Deep-learning-based analysis of an ultrahigh spectral resolution of 2D FT-ICR MSI of prostate cancer tissue. a** Distribution of the optimization convergence with the number of iterations (epochs). **b** The TIC-normalized distribution of original and predicted average spectra. **c** Five-dimensional-encoded features (latent variable z) represent the learned nonlinear manifold that enabled visualization and captured molecular patterns of the original high-dimensional MSI data of 61343 dimensions. These encoded features are of high quality as it were used to predict the original data with an overall mean squared error of $2.42 \times 10^{-5}$. **d** Spatial distribution of a few arbitrary $m/z$ peaks for original and predicted MSI data and it reveals high estimation quality of the original observed data. Comparison between H&E stained histology and clustered molecular patterns reveals molecular-based tumor region: **e** Histopathological annotation of the tumor regions, **f** Gaussian-mixture model (with k = 6) was applied to cluster the encoded features, and the tumor-associated pattern is represented by the light-blue structure (cluster#1) that was extracted (**g**) and correlated with the reduced MSI data. The highest Pearson correlation value was with the ion feature at $m/z$ 739.4664 ± 0.001 and it reveals elevation in the tumor region (**h**).

solely in the latent space of five dimensions. The generative model estimated the observed spectral data given only these five-dimensional encoded features, with an overall reconstruction loss of mean squared error (MSE) of $2.42 \times 10^{-5}$ between *TIC*-normalized spectra of original and predicted data. To visually communicate the reconstruction quality of the MSI data, spatial distributions of a few selected $m/z$ ions of both original and

predicted data are presented in Fig. 2d. The encoded features were then linked to the original $m/z$ variables, using Eq. (4) with setting parameter $\beta$ to 2.5. A reduced list of 244 $m/z$ values revealed the main determinants of molecular patterns captured in the latent space (Supplementary Data 1).

**Identification of molecular patterns associated with tumor regions in human prostate tissue.** The histopathological annotation of the prostate tumor regions revealed a Gleason score (GS) of $(3 + 4) = 7$ (with cribriform cell morphology) (Fig. 2e)[39]. The understanding of molecular patterns underlying the annotated histopathological tumor region could contribute to the development of molecular diagnostics. The encoded features were clustered by the Gaussian-mixture model (GMM) with $k$-clusters ($k = 6$) (Fig. 2f) where the light-blue structure (cluster#1) represents a molecular-based tumor pattern with concordance to the histologically annotated tumor regions. This molecular-based tumor cluster was segmented (Fig. 2g) and correlated with the reduced MSI data of 244 $m/z$ values. For example, the ion feature $m/z$ 739.4664 ± 0.001 with a Pearson correlation coefficient of 0.7 was tentatively assigned to $C_{39}H_{73}O_8P$ by searching the Human Metabolome Database (HMDB)[40] based on the accurate mass and with a tolerance window of 1.44 ppm, $m/z$ 985.5567 ± 0.001 with a correlation coefficient of 0.65 was tentatively identified as PIP(P-42:6) with an error of −0.14 ppm, and $m/z$ 738.4548 ± 0.001 with a correlation coefficient of 0.64 was tentatively identified as PI-Cer(t30:2) with an error of 0.53 ppm. A list of the top determinant $m/z$ values for this tumor cluster with tentative molecular assignments are presented in Supplementary Table 1.

**Tumor-specific metabolic signatures identified in a PDX mouse brain model of glioblastoma.** Four consecutive tissue sections of 12 μm thickness were sampled from a patient-derived xenograft (PDX) mouse brain model of glioblastoma (GBM12) and analyzed by MALDI FT-ICR MSI. The original MSI data was highly sparse and constituted of 3,570 spectra each of 661,402 $m/z$ bins, which was reduced to 21,241 $m/z$ values as presented above. The unsupervised learning process reached stable convergence within less than 100 iterations with a computational time of about 3.6 min (Fig. 3a). The original data were predicted with an overall mean squared error of $4.5 \times 10^{-4}$ (Fig. 3b, e). The five-dimensional encoded features shown in Fig. 3c capture molecular structures located at a nonlinear manifold in the original high-dimensional space. These encoded features were clustered using GMM (k = 8) and the clustered image (Fig. 3d) reveals molecularly distinct tissue regions such as heterogenous tumor regions (cluster#2 and cluster#8) and a tumor rim (cluster#4). Figure 3e shows spatial distribution of some $m/z$ values that were determinant of some molecular clusters. The EGFR inhibitor erlotinib ($m/z$ 394.1757 ± 0.001) and $m/z$ 529.9846 ± 0.001 (tentatively identified as ATP/dGTP with an error of 0.69 ppm) show colocalization with tumor cluster#2, whereas $m/z$ 558.2953 ± 0.001 (tentatively identified as lysoPC(18:2) with an error of 0.62 ppm) is found as part of a second tumor cluster#8. The tumor rim cluster#4 was defined in part by $m/z$ 438.2978 ± 0.001 (tentatively assigned to palmitoylcarnitine with an error of 0.27 ppm). There is a noticeable inverse relationship between the intensity distributions of ATP/dGTP and palmitoylcarnitine within the tumor region. The increased distribution of the palmitoylcarnitine at the interface between normal and tumor tissues was discussed in more details by Randall et al.[41]. Analysis of the neural-network weight variable using Eq. (4) and setting parameter $\beta$ to 2.5 enabled identification of 186 $m/z$ values from the original 21,241 $m/z$ values. Interestingly, $m/z$ 394.175 ± 0.001, which corresponds to the EGFR inhibitor erlotinib was identified despite a mean peak intensity of less than 1%

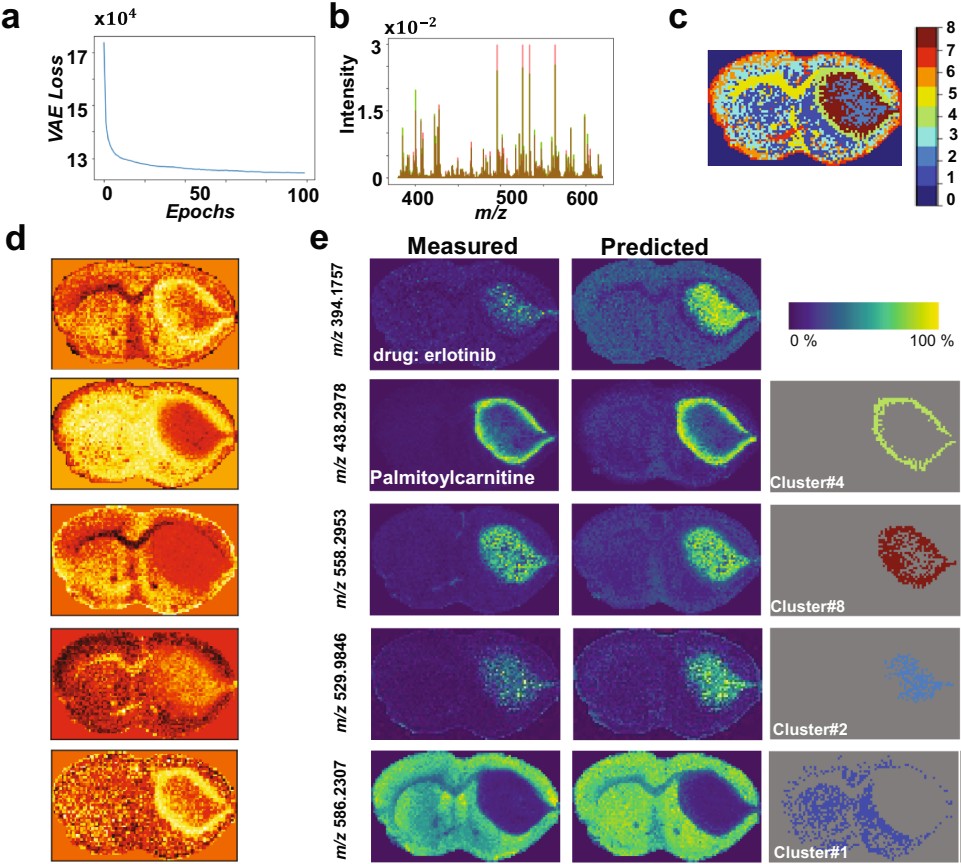

**Fig. 3 Analysis of 2D MALDI FT-ICR MSI dataset of PDX mouse brain model of glioblastoma. a** Distribution of optimization convergence, and **b** Overlay of the mean spectrum of both TIC-normalized original (green) and predicted (red) data with an overall mean squared error of $4.5 \times 10^{-4}$. **c** The clustered image of the encoded features (**d**) using GMM (k = 8) reveals biologically interesting tissue types, such as: normal tissue (cluster#1), tumor heterogeneity (cluster#2 and cluster#8) and a rim around the tumor (cluster#4). **e** Spatial distribution of a few biologically interesting $m/z$ ions that were found highly colocalized within the clusters of interest, and there is a close similarity between the predicted and measured m/z ions.

the value of the largest peak from the mean spectrum. For each of the identified tumor clusters, a list of the top determinant $m/z$ values with tentative molecular assignments are presented in Supplementary Tables 2–4.

**Rapid analysis of 3D MSI data with ultrahigh spectral resolution from a PDX mouse brain model of glioblastoma**. The trained VAE model presented in Fig. 3 was used to analyze an unseen 3D FT-ICR MSI dataset acquired from three consecutive tissue sections (12 μm thickness) with a separation distance of 160 μm of a PDX mouse brain model (GBM12). The dataset was constituted of 11,263 spectra each of 661,402 $m/z$ bins, which was reduced to 21,241 $m/z$ values as presented above and was analyzed using the trained VAE model with a total running time of 6 s for the probabilistic encoder to capture latent variables (encoded features) and about 8 s for the generative model to reconstruct the original spectral data. As shown in Fig. 4a, the original spectral data was predicted with an overall mean squared error of less than $4.12 \times 10^{-4}$. The captured encoded features act as a nonlinear embedding of the high-dimensional data and reveal distinct molecular patterns in the lower-dimensional space of five dimensions (Fig. 4b). Unsupervised clustering of these encoded features using GMM (k = 11) (Fig. 4c) revealed biologically relevant clusters of two heterogenous regions in the tumor core (clusters# 4 and 11) and a tumor rim (cluster#8). Analysis of the neural-network weight variable, using Eq. (4) with setting parameter β to 2.5, enabled identification of 198 $m/z$ values from

the original 21,241 $m/z$ values. The reduced peak list was correlated with the identified biologically relevant clusters and ion features at $m/z$ 438.2978 ± 0.001, 558.2953 ± 0.001, and 529.9846 ± 0.001, which were found to be distributed in the tumor rim region and each of the two heterogenous regions within the tumor core, respectively. Tentative molecular assignment of these $m/z$ values is presented in Supplementary Tables 2–4.

**Scalability on 3D MALDI MSI dataset**. A 79 gigabytes volumetric MSI sample of a mouse kidney that encompasses 73 consecutive tissue sections, each with a thickness of 3.5 μm, was acquired by MALDI MSI in the mass range $m/z$ 2000–20,000 and yielded a 3D spectral image that encompasses 1,362,830 spectra each of 7671 dimensions (i.e., $m/z$ bins)[27]. Of note, this public dataset is available without prior peak picking but underwent some conventional spectral preprocessing such as Gaussian spectral smoothing and baseline subtraction[27]. The computational model was first trained on 18,536 spectra from a 2D MALDI MSI acquisition (section #1), and then tested on the 3D MSI data of the withheld 72 tissue sections (1, 342, 294 spectra). The unsupervised learning process of the training model reached convergence after less than 100 iterations with a total running time of ~8.6 min (Supplementary Fig. 2a) and provided a reconstruction of the original TIC-normalized spectra with an overall mean squared error of $5.5 \times 10^{-3}$ (Supplementary Fig. 2b–d). The five-dimensional encoded features revealed

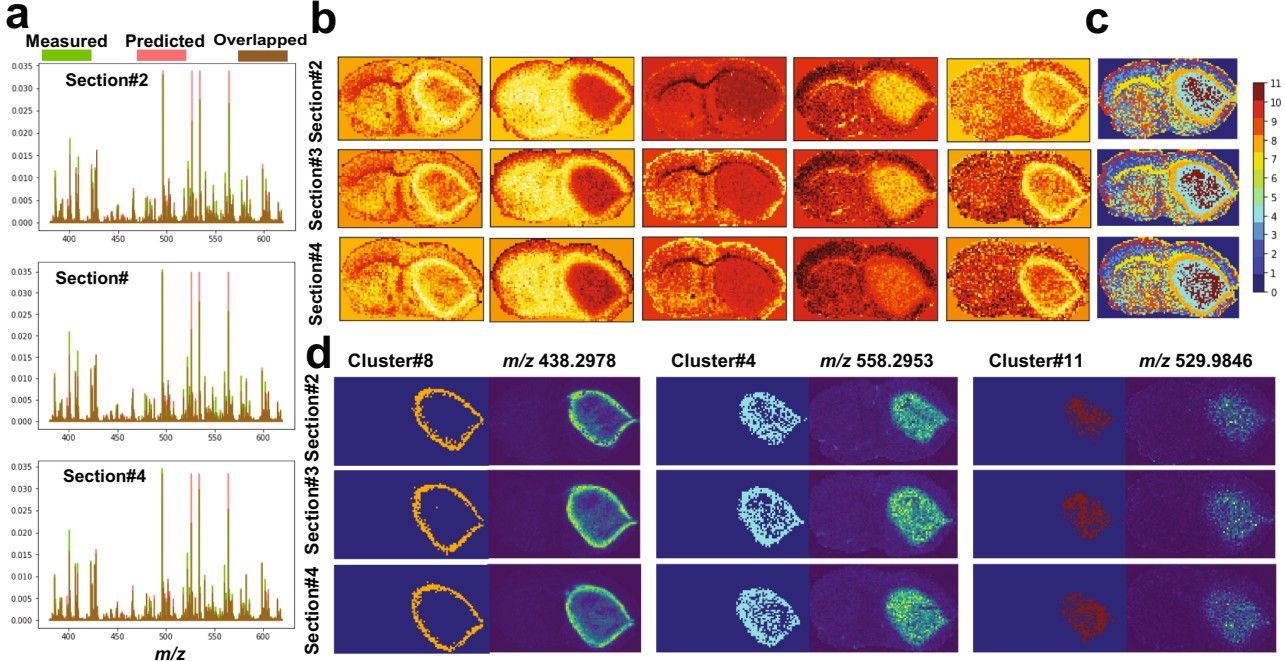

**Fig. 4 Analysis of 3D MALDI FT-ICR test dataset of a PDX mouse brain of glioblastoma.** Analysis on test dataset of MALDI FT-ICR of three consecutive section from a PDX mouse brain of glioblastoma based on the trained model shown in Fig. 3. The analysis was performed independently on each tissue section to reveal: **a** overlay of the overall mean spectrum of both TIC-normalized original (green) and predicted (red) data, **b** five-dimensional encoded features capture molecular structures located on a nonlinear manifold in the original high-dimensional data, **c** GMM-based clustering (k = 11) of the encoded features reveal biologically interesting clusters such as: tumor heterogenous regions (cluster4 and cluster#11) and a rim around the tumor (cluster#8), and **d** spatial distribution of a few learned m/z peaks that were found highly colocalized within distinct tumor clusters.

structural information with distinct molecular patterns (Supplementary Fig. 2e) that were clustered using GMM (k = 7) as shown in Fig. 5f. The molecularly clustered image revealed anatomical structures of the kidney such as the renal cortex (clusters# 2&6), renal medulla (cluster# 1), renal pelvis (cluster# 7), renal artery and vein (cluster# 4), and ureter (cluster# 5) in accordance with histology and previous studies[8,42]. Supplementary Figure 2h shows the spatial distribution of selected m/z values found as part of these molecular patterns. The highly weighted spectral features that constitute the molecular patterns captured by encoded features were identified using Eq. (4) with setting parameter $\beta$ to 2.5 and highlighted in the mean spectrum with m/z bins highlighted in red and resulting peaks highlighted in green (Supplementary Fig. 2g). The complete peak list with 124 m/z values is provided in the Supplementary Data 1.

The trained model was then applied on the 3D MSI data of the withheld 72 tissue sections with spectra corresponding to each 2D MSI tissue section independently loaded into the RAM and analyzed by the trained probabilistic model with an overall running time of about 10 s. The approach was four times faster with 20 times less memory requirement compared to previously reported computational development used to analyze the same dataset[8]. Figure 5 shows results from selected test samples at different volumetric depth within the tissue volume, and results from the analysis of the complete test data are presented in Supplementary Fig. 3. The low-dimensional latent space captured molecular patterns from the high-dimensional space (Fig. 5b) and the encoded features of the entire 3D MSI data were clustered using the GMM (k = 8) revealing molecular patterns that highlight anatomical structures of the kidney (Fig. 5c and Supplementary Fig. 3). The original TIC-normalized MSI data were predicted with an overall mean squared error of $3.11 \times 10^{-3}$.

**Identification of tumor and connective tissue types in 3D DESI MSI of colorectal adenocarcinoma dataset.** DESI MSI data was acquired from 26 consecutive (acquired at every 100 μm) 10 μm thickness tissue sections to reconstruct a 3D MSI volume from a human colorectal adenocarcinoma specimen in the mass range m/z 200–1,050[27]. The 148,044 spectra each of 8,073 dimensions constituting the 3D DESI MSI data volume was analyzed using msiPL. Data from a single tissue section were arbitrary selected (section#1) to train the model with 5,694 spectra, and the model converged after tens of iterations (Supplementary Fig. 3a) with a total running time of ~3.2 min. Based on the learned latent variables, the original spectral data was predicted with an overall mean squared error of $2.02 \times 10^{-4}$ (Supplementary Fig. 4) and the encoded features were then linked to the original m/z values using Eq. (4) with parameter $\beta$ set to 2.5 resulting in a reduced peak list with 24 m/z values. The encoded features were then clustered by GMM (k = 5), revealing a tumor region (red cluster#5) and connective tissue (light-blue cluster#2) in agreement with histological evaluation (Supplementary Fig. 4). These two clusters were correlated with the reduced peak list revealed ions at m/z 279.2 ± 0.1 and m/z 421.3 ± 0.1 that were found to be elevated in the tumor and connective tissue clusters with Pearson correlation values of 0.773 and 0.574, respectively.

The trained model was then applied to analyze the 3D MSI data volume from the remaining 25 tissue sections with 142,350 spectra. To gain memory advantages, the spectra corresponding to each 2D MSI dataset were independently loaded into the memory and analyzed by the VAE model with a running time of approximately 2 s for the probabilistic encoder to predict the encoded features and another 2 s for the generative model to reconstruct the data. The 3D MSI data was predicted with an overall mean squared error of $1.77 \times 10^{-4}$ (Supplementary Figs. 4 and 6) and the 3D encoded features are presented in

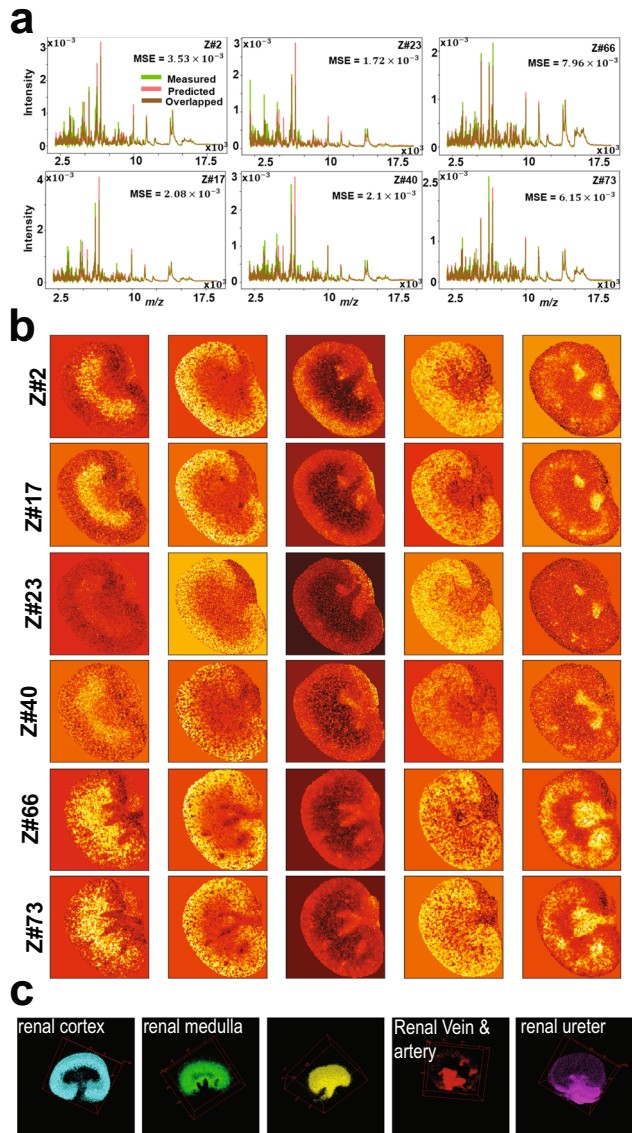

**Fig. 5 Analysis of 3D MALDI MSI test dataset of mouse kidney.** Analysis of 3D MALDI MSI test dataset of mouse kidney of total 72 tissue sections in which spectra of each 2D MSI dataset were independently analyzed: **a** distribution of average spectrum of both TIC-normalized original and predicted data of six datasets samples at different volumetric tissue depth (z-direction). **b** Low dimensional encoded features capture molecular structures from original high-dimensional data. **c** 3D distribution of distinct clusters in the entire dataset that were identified by clustering the encoded features of the entire dataset using GMM (k = 8) and each cluster represents a molecular pattern that reconciles the kidney's anatomy.

Supplementary Fig. 5, representing a five-dimensional nonlinear embedding of the original high-dimensional data. Further GMM clustering (k = 5) of these encoded features revealed two distinct molecular clusters associated with tumor and connective tissues in accordance with previous studies[8,27] and images of the H&E stained tissue sections (Supplementary Fig. 6).

**Identification of α-defensins in human oral squamous cell carcinoma by 3D MALDI MSI data analysis.** The deep learning data analysis strategy was further applied to a 3D MALDI MSI data volume acquired from 58 consecutive tissue sections (10 μm thickness) of a human oral squamous cell carcinoma (OSCC) specimen. Data were acquired in the mass range m/z 2000–20,000

and resulted in a total of 825,558 of preprocessed spectra (Gaussian spectral smoothing and baseline subtraction) each of 7,665 dimensions[27]. The computational model was first trained on a single 2D MALDI MSI dataset (12,875 spectra) of an arbitrary chosen tissue section (section #1), and then tested on MSI data from the withheld 57 tissue sections (815,683 spectra). The training phase reached a stable convergence after less than 100 iterations (Supplementary Fig. 7a), with a total running time of approximately 6.1 min. The original spectral data were predicted with an overall mean squared error of $3.7 \times 10^{-3}$ (Supplementary Fig. 7). The captured encoded features were clustered by GMM (K = 7) and revealed clusters (Supplementary Fig. 7) with underlying distribution of peptide ions at m/z 3,445 and 3,488 found to be elevated (Pearson correlation coefficient of 0.713) with cluster#3. These peptides have previously been proposed to be putative defensins HNP1-3 produced by neutrophils, which can induce tumor angiogenesis[27,43]. Analysis of the neural-network weight variable using Eq. (4) with parameter β set to 2.5 enabled the identification of 177 m/z bins (red markers in Supplementary Fig. 7f) that contributed in forming the molecular patterns captured by the encoded features, leading to a reduced peak list of 44 m/z values (green markers in Supplementary Fig. 7f; Supplementary Data 1).

Spectra from each of the 2D MSI datasets from the remaining 57 tissue sections were independently loaded into the RAM and analyzed by the VAE with an average running time of ~8 s. The proposed approach was 4 times faster and used 30 times less memory compared to reported analysis of the same dataset[8]. Results from a subset 2D MSI datasets taken at different volumetric depth within the tissue volume are presented in Supplementary Fig. 8 with results from the complete data volume (Supplementary Fig. 9). The original TIC-normalized MSI data was predicted with an overall mean squared error of $3.03 \times 10^{-3}$ (Supplementary Fig. 8a) and the encoded features captured molecular patterns that were clustered by GMM (k = 7). Cluster#2 (Supplementary Fig. 8e) highlighted a molecular structure in which lipid ions at m/z 3373, 3445, and 3488 were colocalized and elevated (Supplementary Fig. 8f) as described above for the 2D MSI data analysis and in agreement with a previous study[8]. A reduced list of 57 m/z values strongly correlated to the identified patterns (Supplementary Fig. 8) and results from the complete 57 tissue section volume are shown in Supplementary Fig. 9.

## Discussion

We have proposed msiPL, a generic neural-network-based method for the analysis and peak learning of MSI data from different types of mass spectrometer and tissue type. The neural network showed stability and provided time and computation efficient analysis of various types of MSI data (see Table 1 and Supplementary Table 7). The regularization provided by both the KL-divergence and batch normalization resulted in the stability of the neural network and minimized its reliance on optimization of the hyperparameter initialization[32]. The KL-divergence is embedded in the VAE loss function[29], and the batch normalization was incorporated into the proposed network to normalize each layer's input right before any neuron activation to correct for the co-variate shift[32]. Of note, the number of latent space dimensions is a tunable parameter, which was here empirically set to five dimensions. Generally, a main consideration for choosing the number of latent space dimensions is to minimize the reconstruction error of the generative model through optimizing the VAE cost function shown in Eq. (3). For example, supplementary Table 5 shows the effect of different model parameters on the model complexity and the quality of data reconstruction.

**Table 1 Nonlinear algorithmic performance of MSI spectral data (time and memory comparison).**

| Dataset | | umap | HSNE | msiPL |
|---|---|---|---|---|
| 3D MALDI MSI of Mouse Kidney #Spec = 1,362,830 #$m/z$ = 7671 | Memory/Time | Computationally intractable | 121.8 GB RAM/43 min | <6 GB RAM/ Training: 8.6 min Testing: 10 sec/tissue |
| 3D MALDI MSI of Human OSCC #Spec = 828,558 #$m/z$ = 7665 | Memory/Time | 155 GB RAM/99.95 min | 90 GB RAM/25 min | <6 GB RAM/ Training: 6.1 min Testing: 8 sec/tissue |

Those five-dimensional encoded features are a compressed representation capturing molecular patterns from original high-dimensional data, with each encoded feature combining a set of spectral features rather than a single spectral feature. The link between the encoded features and the original $m/z$ variables was established through backpropagated threshold-based analysis on the neural-network weight matrix. The same threshold weight was given to each encoded feature as indicated by the parameter $\beta$ as defined in Eq. (4). Since some encoded features could capture more significant molecular patterns than other encoded features, the $\beta$ parameter can therefore be optimized for each of those encoded features, which would require further investigation.

The performance of the network was evaluated using five different MSI datasets including 2D MALDI FT-ICR MSI data from human prostate cancer (exported in an imzML file of 9.4 gigabytes of spectral data), 3D DESI MSI data from human colorectal adenocarcinoma (8.9 gigabytes of spectral data), 3D MALDI MSI data from human oral squamous cell carcinoma (47.3 gigabytes of spectral data), and 3D MALDI MSI data from a mouse kidney (78.9 gigabytes of spectral data), and 3D MALDI FT-ICR MSI data from PDX mouse brain model of glioblastoma (3.3 gigabytes of spectral data).

Of note, prior to utilizing the msiPL processing the MSI data was normalized based on the common method of total-ion-count (TIC). However, there are different normalization strategies for MSI data, for example such as those covered i n ref. [44]. It is up to the end users to choose their best MSI normalization strategy; however, the msiPL expects the normalized input data to be bounded within the interval [0,1]. That is because the output layer of the msiPL is based on a sigmoid activation function, which yields values within the range [0,1]. The consistency in the dynamic range at both the input and output layers is crucial to properly optimize the VAE loss function shown in Eq. (3) and eventually minimizing the reconstruction error. The reconstruction error of msiPL was further compared to other methods previously applied on MSI data[19] such as PCA, memory efficient PCA, and Discrete wavelet transform (DWT) followed by PCA. see Supplementary Table 8.

The proposed computational model is set to be trained on the spectral level without considering spatial information with each pixel providing a spectrum as part of a training sample. Individual spectra should be aligned to a common reference, independently of their spatial organization with a 2D or 3D image. The neural network also features mini-batch implementation for parallel processing and for improving memory complexity as it only needs to load a small subset of spectral data into the memory allowing to process large and complex data such as FT-ICR MSI or 3D MSI. This provides an advantage over computational approaches that require the full data loaded into memory to calculate pairwise similarities between spectra[8,42] (see Table 1), allowing the msiPL to be trained on a workstation with a RAM capacity of 32–64 GB. However, and due to the complexity of the neural network that involves optimizing millions of parameters, training the msiPL on the GPU would significantly improve the running time compared to utilizing only the CPU as

demonstrated in Supplementary Table 6. The proposed development provides computational boost and memory advantages that could support a wide variety of MSI applications.

The computational performance of msiPL was compared to two methods for nonlinear dimensionality reduction, namely: 1-Uniform Manifold Approximation and Projection (umap)[45], and 2-Hierarchical Stochastic Neighbor Embedding (HSNE)[26], which is the scalable version of t-SNE[25]. These methods have been used to analyze different types of high-dimensional data[46–49] and a comparison of msiPL to these methods is presented in Table 1 showing improved computational performance for the analysis of large scale MSI data.

The proposed strategy for 3D MSI data analysis is based on a training/testing framework. The training phase runs on a subset of spectra from MSI data from either a single or several tissue sections. The training constitutes of unsupervised learning of the underlying spectral manifold that can subsequently be clustered to identify molecularly distinct regions. The testing phase is applied on the withheld data that do not necessarily need to be fully loaded into the memory at once since the computational model can analyze a subset and capture its underlying molecular patterns in few seconds. This new data analysis approach enabled the overall analysis of 3D MSI data with 20 times less memory and reduced computational time in comparison to umap and HSNE[48], see Table 1. Once identified, the molecular patterns from distinct tissue sections in the volumetric specimen can be reconstructed to form a 3D volume representing the specimen. As an interesting observation, the neural network was able to identify, on test data, spectral patterns of different morphological appearance but similar molecular phenotypes to those encompassed by the training data (e.g., see artery and vein structures in the encoded features of Supplementary Fig. 3 and Fig. 5). It is natural to observe such variations in the morphological phenotype of the tissue anatomy at different locations within the volumetric specimen, but their spectral phenotype should be similar. Therefore, if a certain spectral phenotype was not presented in the training data it would probably not be detected during testing. It should be taken into consideration to balance the spectral phenotypes held in both training and testing datasets, for instance through cross-validation[50] as shown in Fig. 6 and Supplementary Fig. 10, especially in 3D MSI data that may expose molecular heterogeneity within the volumetric specimen.

The stability performance of the computational model was investigated using five-fold cross-validation on the 3D MALDI MSI dataset of mouse kidney as shown in Fig. 6 and Supplementary Fig. 10. The full 3D MSI dataset was randomly shuffled and split into a 20% training set (spanning spectra from 14 tissue sections) and 80% testing set (spanning spectra from 58 tissue sections), see Fig. 6a. For each of the five cross-validation iterations, the msiPL method was applied on the training set to optimize the artificial neural network and the optimized model was then applied on the unseen test set. These analyses revealed comparable performance on both the training and testing sets, as shown in Supplementary Fig. 10. The best cross-validation model was able to reconstruct the original MSI dataset with minimal

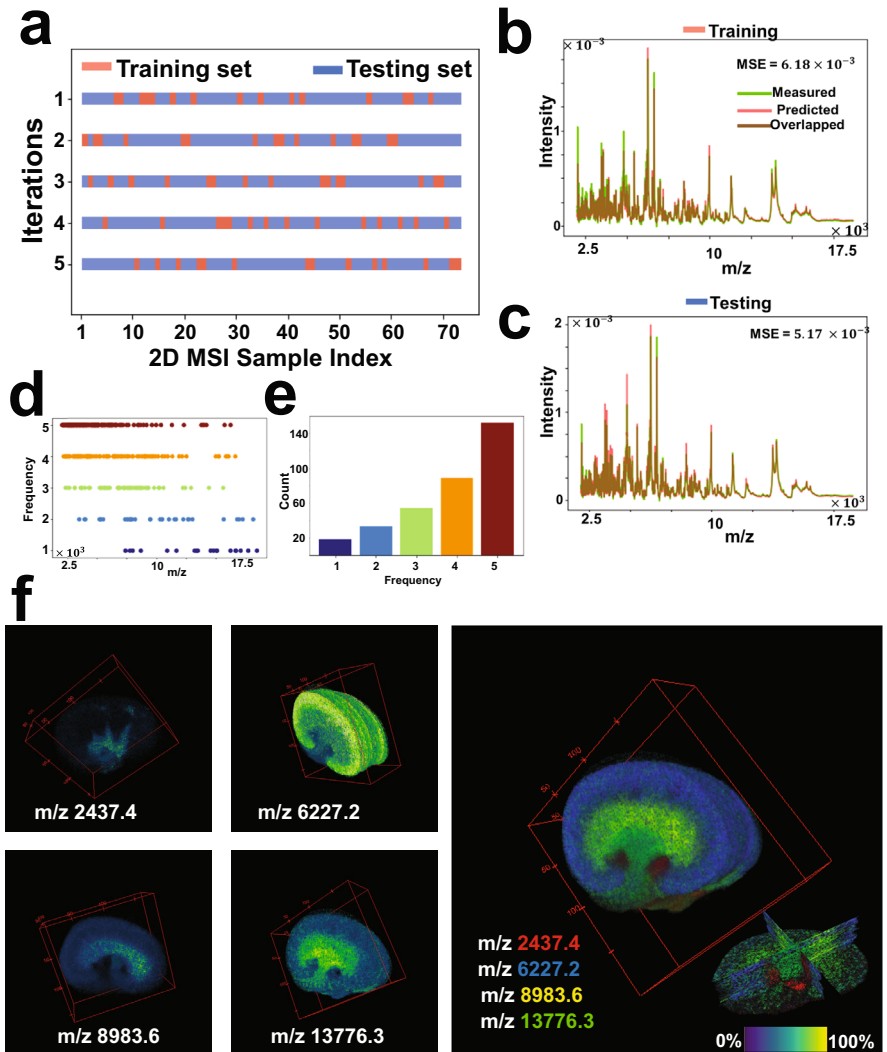

**Fig. 6 Cross-validation analysis for the 3D MALDI MSI data of mouse kidney. a** The full MSI dataset (73 consecutive sections) was randomly shuffled and split into a 20% training set and an 80% testing set, and this process was repeated five times as such for each time the msiPL model was applied on the training set to optimize the artificial neural network and the trained model was then applied on the unseen test set. **b** The best cross-validation model was able to predict the original associated training dataset with minimal mean squared error of $6.18 \times 10^{-3}$, and showing close distribution of their average TIC-normalized spectra. **c** The trained model was applied on the unseen test set and revealed comparable performance. The stability of peak learning across different cross-validation models is with the frequency distribution of all $m/z$ peaks identified in the five-fold cross-validation analyses (**d**), and the peaks count for each frequency (**e**). Overall, 69.6% of the peaks were found stable as they were consistently identified in 80% of the cross-validation analyses. **f** 3D Spatial distribution of selected stable $m/z$ values and each of which reveals high localization to a specific structure that reconciles with the kidney's anatomy, thereby reflecting relevance of the learned peaks.

mean squared error of $6.18 \times 10^{-3}$ and $5.17 \times 10^{-3}$ for the associated training and testing sets, respectively, as given in Fig. 6b, c. The stability of peak learning was investigated across the different cross-validation models. Figure 6d. shows the frequency distribution of all $m/z$ peaks identified in the five-fold cross-validation analyses and the peak count for each frequency is given in Fig. 6e. Overall, 69.6% of the $m/z$ peaks were found stable as they were consistently identified in 80% of the cross-validation analyses. Figure 6f shows 3D spatial distribution of some of those stable $m/z$ peaks and each of which reveals high colocalization to a specific structure that reconciles the kidney's anatomy—reflecting quality of the learned peaks. The full peak list from the different cross-validation models is provided in the Supplementary Data 2. Similarly, another cross-validation analysis was applied on the 3D DESI MSI dataset of colorectal carcinoma and it revealed stable peaks such as $m/z$ 279.2 and $m/z$ 766.5, which

are localized and elevated in the tumor and normal tissue clusters, respectively, as shown in the Fig. 7.

While the neuron activation function of rectified linear unit (ReLU) was used in all layers of the proposed deep learning network, it was not used in the output decoder layer in which the sigmoid function was employed instead. The main reason is that the sigmoid function at the output layer is more adequate for the underlying VAE loss function. The VAE loss function, as illustrated in Eq. (3) consists of summation of two terms of KL-divergence and the marginal likelihood estimate that was modeled using categorical cross-entropy. The KL-divergence acts as a regularizer for the probabilistic encoder to measure the similarity between the approximate estimate distribution $q_\phi(z|x)$ and the true but intractable distribution $p_\Theta(z|x)$. The categorical cross-entropy was used to measure the reconstruction loss between two probability distributions of original input and the estimated marginal likelihood

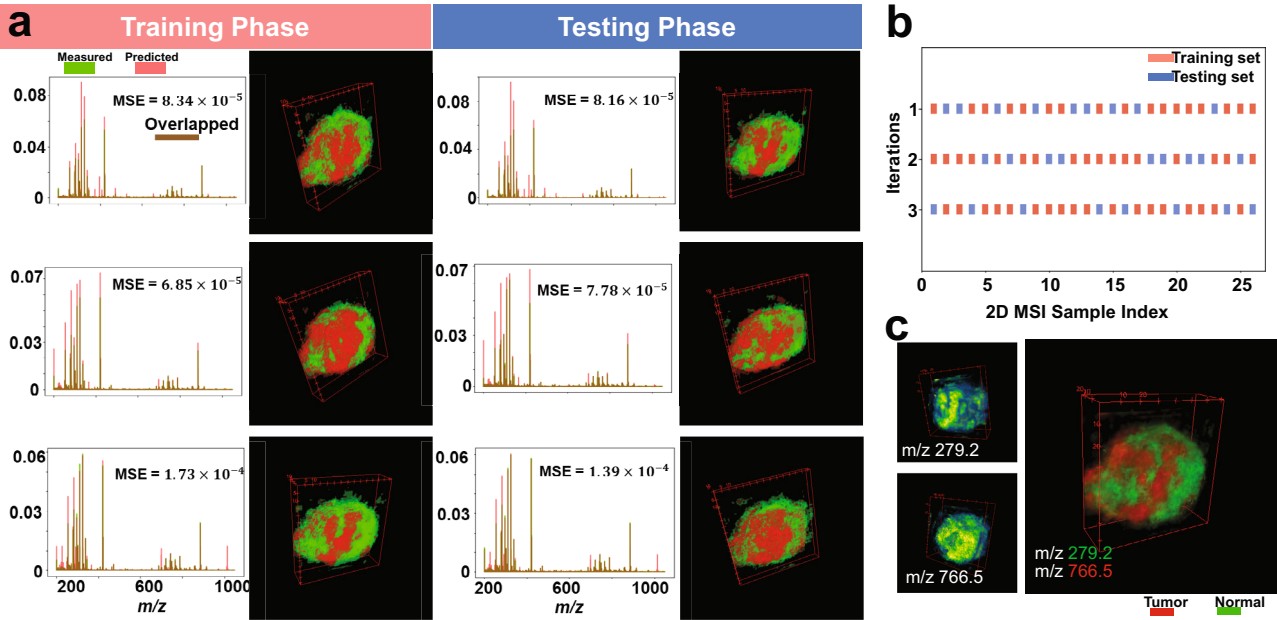

**Fig. 7 Cross-validation analysis for the 3D DESI MSI dataset of a human specimen of colorectal carcinoma. a** The training set was used to optimize the neural network and then the trained model was applied on the testing set, and this process was repeated three times (rows) according to three-fold cross-validation shown in (**b**) in which the full dataset was randomly shuffled and split into training and testing sets. There is a close consensus in the performance of the cross-validated models in predicting the original data, learning the nonlinear manifold, and identifying the tumor and normal clusters. **c** The three cross-validated models showed stability in learning peaks of interest such as *m/z* 279.2 and *m/z* 766.5 that were found localized (>0.7 Pearson correlation) and elevated in the tumor and normal clusters, respectively.

represented by the decoded output. Unlike the ReLU activation function, the sigmoid function output can be interpreted as a probability since its range varies between 0 and 1[51].

The peak learning performance of msiPL was benchmarked against a classical approach by performing peak picking on a mean spectrum. The classical approach is based on the orthogonal matching pursuit (OMP) algorithm[52,53], which is implemented in the software SCiLS lab (version 2020a, Bruker). The OMP algorithm for peak picking was applied on the mean spectrum of the 3D MALDI MSI PDX GBM dataset using the software's default values. The peaks identified by both msiPL and the OMP are highlighted in Supplementary Fig. 11a, and listed in the Supplementary Data 3. While 38% of the peaks were picked by both methods, the msiPL method exclusively picked 53% of the total identified peaks whereas 9% of the peaks were exclusively picked by the classical approach (Supplementary Fig. 11). The performance of msiPL in identifying peaks of lower intensities, as shown in Supplementary Fig. 11b, c, is attributed to the underlying employed concept of manifold learning that focuses on learning *m/z* patterns regardless of peak intensity and shape. The majority of peaks identified by msiPL in the 3D MSI PDX GBM dataset (Supplementary Tables 2–4) were missed by the classical OMP method applied to the mean spectrum for rapid analysis (few seconds). The sensitivity for peak identification with such a classical method could be improved by optimizing peak picking parameters[15] and by analyzing a wider number of spectra[17,41], which results in longer processing time (Supplementary Table 7). Moreover, a recent study by Murta et al. showed that the selection of peak picking parameters does not only affect the clustering analysis but could in turn impact biological interpretations[54].

Analysis of mass spectrometry imaging data with bypassing conventional spectral preprocessing would enrich the biomarker discovery process by increasing identification sensitivity and specificity for molecular ions with biological relevance. For

instance, this would minimize the effect of optimization in pre-processing steps required for feature extraction such as peak picking that have an impact on overall biomarker identification. Despite manufacturers currently offering default parameter settings for the peak picking process (e.g., spectral smoothing, signal-to-noise ratio (S/N), peak shape, and peak threshold), these default values are general and may lack the sensitivity needed to identify peaks of lower intensity[15] (e.g., Supplementary Fig. 11). Additionally, the quadratic computational complexity of the analysis slows down the peak picking process with an increase in the number of spectra[17], see Supplementary Table 7. Of note, the local maxima detection applied here to reduce the original FT-ICR MSI data complexity conceptually differs from peak picking. The former identifies local maxima between three consecutive *m/z* variables[13,17], while the latter seeks to identify relevant peaks induced by molecules which are spectrally characterized by a structure that includes (peak height, width, slope, and baseline) and minimize the contribution of noise-to-signal ratio. However, the local maxima approximation applied to the FT-ICR MSI data could be avoided in future developments through, for example, investigating an integrated multi-spectral scale neural-network architecture. Such future work would allow to cope with even more complex spectral data such as those acquired by ultrahigh resolution FT-ICR MSI instruments[55]. Future developments may also consider another level of spectral complexity such that provided by the collision cross section property of the trapped ion mobility spectrometry (TIMS) based technology which holds promise for molecular identification[56]. Classical machine learning approaches are of limited capabilities to analyze original mass spectrometry data at full spectral dimensions. Mainly, because those approaches suffer a common issue known as curse-of-dimensionality that deteriorates the clustering/classification accuracy on high-dimensional data[57]. In contrast, deep-learning-based approaches have shown the ability to avoid the curse-of-dimensionality and to establish self-learning of relevant features

that increase classification accuracy[58]. This expansion in data analysis could impact complex applications such as multiclass classification required to resolve molecular intratumor heterogeneity.

The GMM-based clustering method on the encoded features has been found computationally fast (i.e., a few seconds) as well as efficient in identifying spatial clusters of biological relevance such as distinguishing normal and tumor clusters. The number of K-clusters is a user tunable parameter that could be set either manually or automatically. In our strategy, an iterative approach was followed in which the GMM clustering process was applied on a different number of K-clusters within an expected range. The best model was then manually chosen as such it exhibited a clustering balance that avoids under-/overestimation, but an information theory-based optimization process can be utilized for automated model selection, see Supplementary Fig. 12. That is the Bayesian information criterion (BIC) algorithm in which the best model is theoretically expected to achieve the minimal BIC score[59]. Since the distribution of the BIC scores is gradually decreasing within the searchable range, as shown in Supplementary Fig. 12, the Kneedle algorithm[60] (using the python public package "Kneed") was then applied on the BIC scores to detect the critical point of maximum curvature (also known as elbow/knee point) at which the best model was selected. Compared to the manual approach, there was a noticeable underestimation for the PDX GBM dataset in which intratumor heterogeneous clusters were missed whereas an overestimation was observed for the colorectal cancer dataset.

We sought to investigate the capabilities of a trained msiPL model to analyze unseen data of similar tumor type but from a different subject with different tumor grade. Here, the model trained with a MALDI FT-ICR MSI dataset from prostate cancerous tissue with a Gleason score $(3 + 4)$ (Fig. 2) was applied to the analysis of a distinct MALDI FT-ICR MSI dataset, see Supplementary Fig. 13. The histopathological annotation of the test tissue section revealed two cancerous regions with distinct Gleason scores of $(5 + 4)$ and $(3 + 4)$ (Supplementary Fig. 13d). The test MSI dataset, constituting of 13,471 spectra each with 61,343 $m/z$ values, was analyzed by the trained msiPL model in 56 s. The model was able to predict and reconstruct the original test data with an overall mean squared error of $2.273 \times 10^{-5}$, and the overlay of the average TIC-normalized spectrum of both original and predicted data are shown in Supplementary Fig. 13a. The distribution of the GMM model selection criterion based on the Bayesian information criterion (BIC) and Kneedle algorithm revealed an optimal number of K-clusters (K = 11) (Supplementary Fig. 13b), which was applied to a clustering analysis of the encoded features (Supplementary Fig. 13c, e). Of interest, the model captured a spatial cluster associated with the histopathological annotation of Gleason score $(5 + 4)$ (Supplementary Fig. 13f), and the Pearson correlation analysis revealed the highest correlated ion feature at $m/z$ $786.5981 \pm 0.001$ (Supplementary Fig. 13g). In accordance with a recent study that analyzed the same dataset, we noticed clustering distinction between two Gleason scores of $(5 + 4)$ and $(3 + 4)$[61]. While our results support the efficiency of msiPL for unsupervised mining of different MSI datasets and identification of spatial patterns of biological relevance, we envision future extensions of the msiPL model to enable classification and predictive tasks for tumor type and grade directly from the mass spectral data.

Taken together, the deep learning data analysis strategy presented here provides the ability to learn the underlying nonlinear manifold required to identify and visualize molecular patterns from original high-dimensional data, avoiding preprocessing computation. The resulting workflow provides improved data analysis time of large and complex new data, while delivering an enriched biomarker discovery process through unsupervised identification of complex molecular patterns with identification of their determinant $m/z$ values.

## Methods

**Experimental datasets.** MSI datasets from five different tissue types were analyzed and their description is given in the Supplementary Materials and Methods. Briefly, three of these MSI datasets are publicly available[27], which include: 3D DESI MSI dataset of human colorectal adenocarcinoma, 3D MALDI MSI dataset of human oral squamous cell carcinoma, and 3D MALDI MSI data of mouse kidney. The other two MSI datasets were collected and acquired at our institution, which include: 2D MALDI FT-ICR MSI dataset of human prostate cancer[62], and 3D MALDI FT-ICR MSI dataset of PDX mouse brain model of glioblastoma[63]. The MSI datasets, without prior peak picking, were exported in the standardized format imzML[37] using SCiLS Lab (2019c, Bruker) and converted to HDF5 format[38] for variational autoencoder analysis.

**Variational autoencoder.** Mathematically, spectra of MSI data can be represented as a set of high-dimensional vectors $X = \{x^{(1)}, x^{(2)}, \ldots, x^{(N)}\}$, where $N$ represents the total number of spectra (or pixels) and each spectrum $x^{(i)} \epsilon \mathbb{R}^d$ is of $d$-dimensions. We assume these $d$-dimensional i.i.d. vectors were generated by a random process that involves an unobserved lower-dimensional latent variable $z \epsilon \mathbb{R}^k$, where $d \gg k$. The latent variable is sampled from a prior distribution $p_{\Theta*}(z)$ and the datapoint $x^{(i)}$ is generated from the conditional distribution $p_{\Theta*}(x|z)$. The true posterior distribution $p_{\Theta*}(z|x)$ would provide a compressed representation (we referred to it as encoded features) of the observed high-dimensional data. The high-dimensional nature of $x^{(i)}$ makes the posterior distribution computationally intractable. The variational inference aims therefore at introducing a recognition model $q_\phi(z|x)$ that approximates the true intractable posterior $p_\Theta(z|x)$. The recognition model is assumed to be sampled from a normal distribution parameterized by $\mu_\phi(x)$ and $\sigma_\phi(x)$ as shown in Eq. (1). The inference of the latent variable $z$ would enable the generative model of marginal likelihood estimator $p_\Theta(x|z)$ to reconstruct datapoint $x^{(i)}$.

$$q_\phi(z, |, x) \sim \mathcal{N}(\mu_\phi(x), \sigma_\phi(x)\mathbf{I}). \tag{1}$$

The variational parameter $\phi$ needs to be estimated as such it makes $q_\phi(z|x)$ as close as possible to the true posterior $p_\Theta(z|x)$. The Kullback-Leibler (KL) divergence, given in Eq. (2), can assess the closeness between these two distributions. Because the KL-divergence is always non-negative, the term $(E_{q_\phi(z|x)}[\log q_\phi(z|x)] - E_{q_\phi(z|x)}[p_\theta(z, x)] = \mathscr{L})$ represents the variational lower bound on the marginal distribution, where $\mathscr{L} \leq \log p_\Theta(x)$. The optimum estimate of the parameters $\phi$ and $\Theta$ would maximize the variational lower bound that can be rewritten as shown in Eq. (3).

$$\begin{aligned} KL(q_\phi(z|x) \parallel p_\theta(z|x)) &= \int q_\phi(z|x) \log \frac{q_\phi(z|x)}{p_\theta(z|x)} dz \\ &= E_{q_\phi(z|x)}[\log q_\phi(z|x)] - E_{q_\phi(z|x)}[p_\theta(z, x)] + \log p_\theta(x). \end{aligned} \tag{2}$$

In variational autoencoder (VAE), the recognition model represents a probabilistic encoder and the generative model represents a probabilistic decoder. The recognition and generative model parameters $\phi$ and $\Theta$ are computed from the neural-network parameters and jointly optimized by maximizing the cost function of the variational lower bound $\mathscr{L}(\phi, \Theta; x^{(i)})$, given in Eq. (3), which eventually would minimize the overall VAE loss. The first term of this cost function acts as a regularizer for the encoder and it measures the closeness between the approximated posterior and the prior. The second term represents the expected value of the prediction error which we modeled as a cross-entropy.

$$\mathcal{L}(\phi, \theta; x^{(i)}) = -KL\left(q_\phi(z|x^{(i)}) \parallel p_\theta(z)\right) + E_{q_\phi(z|x^{(i)})}[\log p_\theta(x^{(i)}|z)]. \tag{3}$$

Kingma et al. introduced a reparameterization trick to make $\mathscr{L}(\phi, \Theta; x^{(i)})$ differentiable with respect to $\phi$ and $\Theta$. That is, incorporating first an auxiliary variable $\varepsilon \sim \mathcal{N}(0, 1)$ with an input datapoint $x^{(i)}$ to form a continuous function $g_\phi(\varepsilon, x^{(i)})$ that can then be used to sample the latent variable from the approximated posterior $z^{(i)} \sim q_\phi(z, |, x^{(i)})$ as such $z^{(i)} = g_\phi(\varepsilon, x^{(i)}) = \mu_\phi(x^{(i)}) + \text{diag}\left(\sigma_\phi(x^{(i)})\right).\varepsilon$. For more information on variational autoencoder, we refer to ref. [29].

**Identification of Informative $m/z$ peaks.** The inferred multivariate latent variable $z$ represents encoded features that capture molecular patterns in the original MSI data. It is therefore of interest to identify $m/z$ features underlying those learnt patterns. We propose a threshold analysis on the weight parameter $W^{(L)}$ of the neural network at layer $L$, as depicted in Fig. 1d. Briefly, for each encoded feature represented by the $i$th neuron at layer $h_2$ we first identify the $j$th neuron at the previous hidden layer $h_1$ with maximum scaler weight value $w_{ij}^{(2)} \subset W^{(2)}$, see red line in Fig. 1d. Then, a threshold $T$ was computed, Eq. (4), using the weight vector $w_{dj}^{(1)} \subset W^{(1)}$, which is a one-dimensional vector holds the weights between all $d$ neurons of the input layer $L_1$ and the identified $j$th neuron at $h_1$. Eventually, a set of

$p$ neurons at the input layer $L_1$ whose weights are larger than $T$ (i.e., $w_{dj}^{(1)} \geq T$) were retrieved and each of which represents an $m/z$ variable, for schematic illustration see blue highlighted lines in Fig. 1d. Since the original MSI data were analyzed without prior preprocessing for peak picking, the retrieved observed variables represent $m/z$ bins that need then to be assigned to its associated $m/z$ peak. As such, an $m/z$ peak has been identified on the average spectrum as the nearest local maximum to a given $m/z$ bin.

$$T = \text{mean}(w_{dj}^{(1)}) + \beta * \text{std}(w_{dj}^{(1)}); \text{ where } \beta \in [1, 2.5]. \quad (4)$$

**Data clustering using Gaussian Mixture Model (GMM)**. The encoded features reduce the original dimensional complexity and enable application of a simple clustering algorithm such as Gaussian-mixture model (GMM)[64]. The encoded features are expected to learn a nonlinear manifold to allow capturing and visualizing molecular patterns from original high-dimensional data. The clustering algorithm would cluster those patterns to form one image in which distinct clusters represent molecularly distinct regions. The number of clusters (k) for the GMM clustering algorithm is a user tunable parameter that can also be automated using the Bayesian information criterion[59]. A cluster of interest is then correlated with the MSI data of reduced peak list to identify colocalized $m/z$ peaks with the highest Pearson correlation coefficient.

**Reporting summary**. Further information on research design is available in the Nature Research Reporting Summary linked to this article.

## Data availability

These three MSI datasets (3D MALDI TOF MSI data of human oral squamous cell carcinoma, 3D MALDI TOF MSI data of mouse kidney, and 3D DESI Orbitrap MSI data of human colorectal adenocarcinoma) were previously published and publicly available by Oetjen et al.[27]. The 2D MALDI FT-ICR MSI data generated from human prostate tissue, and the 3D MALDI FT-ICR MSI data generated from a PDX glioblastoma mouse model in this study have been deposited in the NIH Common Fund's National Metabolomics Data Repository (NMDR) Metabolomics Workbench (https://www.metabolomicsworkbench.org) under project id (PR001171) with https://doi.org/10.21228/M8BM4Q. The Human Metabolome Database (https://hmdb.ca/) was used for annotation of $m/z$ values.

## Code availability

Source code is publicly available on GitHub: https://github.com/wabdelmoula/msiPL.git.

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

## Acknowledgements

We would like to acknowledge the contribution from Dr. Sylwia Stopka in revising the manuscript. This work was funded by NIH U54 CA210180 MIT/Mayo Physical Science Oncology Center for Drug Distribution and Drug Efficacy in Brain Tumors, and by the Dana-Farber Cancer Institute PLGA Fund. E.C.R. was in receipt of an NIH R25 (R25 CA-89017). NYRA receives support from the Advanced Technologies-National Center for Image Guided Therapy (AT-NCIGT) NIH P41EB028741 and NIH R01CA201469.

## Author contributions

W.M.A., F.M.W., J.N.A. and N.Y.R.A. designed research. W.M.A., B.G.C.L. and E.C.R. performed research and analyzed data. J.N.A. and N.Y.R.A. contributed analytical tools. J.N.S. contributed animal models. W.M.A., T.K., F.M.W., J.N.A., W.M.W. and N.Y.R.A. wrote the paper.

## Competing interests

W.M.A. is now an employee of inviCRO. N.Y.R.A. is a scientific advisor to BayesianDx and inviCRO, and key opinion leader to Bruker Daltonics. The remaining authors declare no competing interests.
