## [Peer Review File · Nature Communications]

REVIEWER COMMENTS

Reviewer #1 (Remarks to the Author):

The manuscript by Abdelmoula et al. presents an approach of using an autoencoder to obtain a low-dimensional latent representation of mass spectrometry imaging (MSI) data.

For full disclosure, I've reviewed the previous versions of the manuscript for Nature Methods and this review integrates my previous reviews, as I found this version of the manuscript very closely resembling the last version I've seen.

The authors show that this approach helps reconstruct data with high precision and is fast. However, the manuscript has several major issues.

First major issue: The manuscript claims that peak picking is a major unsolved challenge in MSI and criticizes some other approaches (without a comprehensive review) of being subjective and requiring user input. It's hard to accept both statements. Several major MSI vendors (Bruker, Thermo) perform peak picking on the fly for their data that requires no user input. This basically removes the need for custom-developed peak picking algorithms. Moreover, the authors fall into the trap of their own criticism by providing yet another subjective method without any objective evaluation. The only comparison presented in Figure S11 which shows one peak that is not detected in a vendor software but was detected by the authors' method. However, showing just one peak from one spectrum cannot be considered as comparison or evaluation. Concerning the speed improvement, Table S6 compares runtimes between the proposed method and a commercial software. The improvement is ~7 times for a regular FTICR dataset and from 10 hours to 4 minutes for a 3D dataset when using an approximate training/testing strategy. However, the authors have compared their method only against one method in vendor software and actually a slow one. If they performed peak picking on a mean spectrum, then the results would be achieved much faster and likely faster than their own method (I cannot reproduce as the data was not provided). Would such a simple but fast method deliver worse results? It's not shown.

Second major issue: The authors provide no comparison with state of the art methods for dimensionality reduction. This is not acceptable for the present state of the field where tens of dimensionality reduction methods were developed; see (Verbeek et al 2019 Mass Spectrom Rev) for a review and a list of existing methods.

Third major issue: The authors provide no objective evaluation of their VAE-based method other than by showing that it is able to reconstruct the data with high precision as well as it is 4 times faster and is more memory efficient than their previous implementation. I speculate that many methods listed in (Verbeek et al 2019 Mass Spectrom Rev) can achieve similar performance.

Fourth major issue: The availability statement "Source code and scripts will be available on GitHub at time of publication." is not acceptable for a manuscript proposing a novel computational method, since it provides no opportunity for reviewers to evaluate the code, its correctness as well as to make sure that the repository covers all aspects necessary for reproducing the results.

Reviewer #2 (Remarks to the Author):

The manuscript by Abdelmoula et al. describes a method (a deep learning tool) that can visualize and reveal biologically-relevant clusters of tumor heterogeneity and identify underlying informative m/z peaks from large data sets acquired by mass spectrometry imaging (MSI). The authors demonstrate the robustness and generic applicability of the method on MSI data of large size from different biological systems acquired using different mass spectrometers. The manuscript is well and clearly presented. The method addresses a key challenge and has a potential for having a broad impact in the field.

1. The performance of the proposed method was tested using various 2D and 3D MSI using different MSI platforms. It is however not clear to reviewer how this method compares with other

methods currently used? The authors mention that the method 'provided time and computation efficient analysis of various types of MSI data'. Compared to which method? How much more efficient? Please include a description about the methods currently used in MSI for these types of studies and include quantitative data.

2. The performance of the method was also tested using various biological samples collected from different organs; i) human prostate cancer tissue specimens, ii) a mouse brain model of glioblastoma, iii) human colorectal adenocarcinoma and iv) human oral squamous cell carcinoma. Molecular patterns associated with tumour regions were identified and ion features were tentatively 'identified' based on accurate mass. The authors tested the method on different tissue types but they did not use the method for defining different grades of e.g., human prostate cancer. Instead of providing data from many different cancer types it would have been much more interesting to test and define the method on different grades of the cancers. Please include an example(s).

RESPONSE TO THE REVIEWER COMMENTS

Reviewer #1 (Remarks to the Author):

The manuscript by Abdelmoula et al. presents an approach of using an autoencoder to obtain a low-dimensional latent representation of mass spectrometry imaging (MSI) data. For full disclosure, I've reviewed the previous versions of the manuscript for Nature Methods and this review integrates my previous reviews, as I found this version of the manuscript very closely resembling the last version I've seen.

The authors show that this approach helps reconstruct data with high precision and is fast. However, the manuscript has several major issues.

First major issue: The manuscript claims that peak picking is a major unsolved challenge in MSI and criticizes some other approaches (without a comprehensive review) of being subjective and requiring user input. It's hard to accept both statements. Several major MSI vendors (Bruker, Thermo) perform peak picking on the fly for their data that requires no user input. This basically removes the need for custom-developed peak picking algorithms. Moreover, the authors fall into the trap of their own criticism by providing yet another subjective method without any objective evaluation. The only comparison presented in Figure S11 which shows one peak that is not detected in a vendor software but was detected by the authors' method. However, showing just one peak from one spectrum cannot be considered as comparison or evaluation. Concerning the speed improvement, Table S6 compares runtimes between the proposed method and a commercial software. The improvement is ~7 times for a regular FTICR dataset and from 10 hours to 4 minutes for a 3D dataset when using an approximate training/testing strategy. However, the authors have compared their method only against one method in vendor software and actually a slow one. If they performed peak picking on a mean spectrum, then the results would be achieved much faster and likely faster than their own method (I cannot reproduce as the data was not provided). Would such a simple but fast method deliver worse results? It's not shown.

Ans: Please note that both data and code were provided. A quantitative comparison of peak learning using msiPL and a classical approach of peak picking on the mean spectrum is now included in the main manuscript and summarized in Figure S11 below which is now included in the supplementary materials. We based the classical peak picking approach on the orthogonal matching pursuit (OMP) algorithm that is implemented in the Bruker software SCiLS lab (version 2020a). The OMP algorithm for peak picking was applied on the mean spectrum of the 3D MALDI MSI PDX GBM dataset using the software's default values. We have then replaced Figure S11 in original submission with the new Figure S11 below. The following text is now added to the main manuscript:

“The peak learning performance of msiPL was benchmarked against a classical approach by performing peak picking on a mean spectrum. The classical approach is based on the orthogonal matching pursuit (OMP) algorithm [Pati et al, IEEE 1993; and Alexandrov et al., J. Proteome Research 2010] which is implemented in the software SCiLS lab (version 2020a, Bruker). The OMP algorithm for peak picking was applied on the mean spectrum of the 3D MALDI MSI PDX GBM dataset using the software's default values. The peaks identified by both msiPL and the OMP are highlighted in Figure S11.a, and listed in the Supplementary File “Peaks_msiPL_OMP(MeanSpec).xlsx”. While 38% of the peaks were picked by both methods, the msiPL method exclusively picked 53% of the total identified peaks whereas 9% of the peaks were exclusively picked by the classical approach (Figure S11). The performance of msiPL in identifying peaks of lower intensities, as shown in Figure S11.b-c, is attributed to the underlying employed concept of manifold learning that focuses on learning m/z patterns regardless of peak intensity and shape. The majority of peaks identified by msiPL in the 3D MSI PDX GBM dataset (Supplementary Tables S2-S4) were missed by the classical OMP method applied to the mean spectrum for rapid analysis (few seconds). The sensitivity for peak identification with such a classical method could be improved by optimizing peak picking parameters [Donnelly et al., Nature Methods 2019] and by analyzing a wider number of spectra [Alexandrov, BMC Bioinformatics. 2012; Randall et al. Cancer Research 2020], but this results in longer processing time (Supplementary Table S7). Moreover, a recent study by Murta et al. showed that the selection of peak picking parameters does not only affect the clustering analysis but could in turn impact biological interpretation [Murta et al. Anal Chem, 2021]. In contrast, msiPL allows rapid and sensitive spectral data processing without the optimization of pre-processing parameters.”

Figure S11. Comparison of peak picking analysis on the 3D MALDI FT-ICR MSI PDX GBM dataset: (a) using msiPL on full spectral data and the orthogonal matching pursuit (OMP) algorithm on the mean spectrum (implemented in the commercial software of SCiLS Lab version 2020a (Bruker, Bremen, Germany)). (b-c) visualization of some of the peaks with biological relevance identified only by the manifold learning approach of msiPL but not by a classical approach of applying the OMP algorithm on the mean spectrum.

Second major issue: The authors provide no comparison with state of the art methods for dimensionality reduction. This is not acceptable for the present state of the field where tens of dimensionality reduction methods were developed; see (Verbeeck et al 2019 Mass Spectrom Rev) for a review and a list of existing methods.

Ans: We have now quantitatively benchmarked the computational performance of msiPL against two dimensionality reduction algorithms namely: 1- Uniform Manifold Approximation and Projection (umap), and 2- Hierarchical Stochastic Neighbor Embedding (HSNE) which is a scalable version of the widely used t-SNE method. The main manuscript has been updated accordingly by including the following Table 1 and red-highlighted text.

“The computational performance of msiPL was compared to two state-of-the-art methods for non-linear dimensionality reduction, namely: 1- Uniform Manifold Approximation and Projection (umap) [McInnes et al. arXiv:1802.03426,2018], and 2- Hierarchical Stochastic Neighbor Embedding (HSNE) [Pezzotti et al. Computer Graphics Forum,2016] which is the scalable version of t-SNE. These methods have been effectively used to analyze different types of high-dimensional data [Becht et al. Nature biotechnology (2019); van Unen et al. Nature communications (2017); Korsunsky et al. Nature Methods (2019)], nevertheless they exhibit computational limitations for large scale MSI data analysis (e.g. MSI data with $10^5 - 10^6$ spectra without prior peak picking). Table 1 shows that the computational performance of msiPL outperforms both umap and HSNE for the analysis of large scale MSI data.”

Table 1. Non-linear algorithmic performance of MSI spectral data (time and memory comparison)

Dataset		umap	HSNE	msiPL
3D MALDI MSI of Mouse Kidney #Spec = 1,362,830 #m/z = 7671	Memory	Computationally Intractable	121.8 GB RAM	<6 GB RAM
	Time		43 min	Training: 8.6 min Testing: 10 sec/tissue
3D MALDI MSI of Human OSCC #Spec= 828,558 #m/z = 7665	Memory	155 GB RAM	90 GB RAM	<6 GB RAM
	Time	99.95 min	25 min	Training: 6.1 min Testing: 8 sec/tissue

Third major issue: The authors provide no objective evaluation of their VAE-based method other than by showing that it is able to reconstruct the data with high precision as well as it is 4 times faster and is more memory efficient than their previous implementation. I speculate that many methods listed in (Verbeeck et al 2019 Mass Spectrom Rev) can achieve similar performance.

Ans: We had originally assessed the learning quality of the VAE model through quantitative assessment using the mean squared error metric between the model’s reconstructed data and the original ground truth. This is an established quantitative machine learning evaluation criterion which is neither subjective nor biased since it is strictly data driven.

Fourth major issue: The availability statement "Source code and scripts will be available on GitHub at time of publication." is not acceptable for a manuscript proposing a novel computational method, since it provides no opportunity for reviewers to evaluate the code, its correctness as well as to make sure that the repository covers all aspects necessary for reproducing the results.

Ans: In the original submission, specifically on page#24 of the manuscript, we had provided the GitHub link to our python code.

Reviewer #2 (Remarks to the Author):

The manuscript by Abdelmoula et al. describes a method (a deep learning tool) that can visualize and reveal biologically-relevant clusters of tumor heterogeneity and identify underlying informative m/z peaks from large data sets acquired by mass spectrometry imaging (MSI). The authors demonstrate the robustness and generic applicability of the method on MSI data of large size from different biological systems acquired using different mass spectrometers. The manuscript is well and clearly presented. The method addresses a key challenge and has a potential for having a broad impact in the field.

Ans: Thank you, we very much appreciate the positive feedback and appreciation of our work in that we are addressing a key challenge in the field.

1. The performance of the proposed method was tested using various 2D and 3D MSI using different MSI platforms. It is however not clear to reviewer how this method compares with other methods currently used? The authors mention that the method 'provided time and computation efficient analysis of various types of MSI data'. Compared to which method? How much more efficient? Please include a description about the methods currently used in MSI for these types of studies and include quantitative data.

Ans: Thank you for raising this interesting point. Based on your request, we have performed a quantitative assessment to benchmark the computational performance of our method compared to the state-of-the-art in non-linear dimensionality reduction used for MSI data analysis. Also, we had previously benchmarked the computational performance of peak picking between our method and the widely used commercial software of SCiLS lab (Bruker, Germany).

We have now directly compared the computational performance of msiPL to two commonly used non-linear algorithms: 1- Uniform Manifold Approximation and Projection (umap) [McInnes et al. arXiv:1802.03426,2018; Becht et al. Nature biotechnology,2019], and 2- Hierarchical Stochastic Neighbor Embedding (HSNE) [Pezzotti et al. Computer Graphics Forum,2016] which is the scalable version of the widely used method of t-SNE. Both umap and HSNE/t-SNE are non-linear dimensionality reduction methods, which have been effectively used over the last decade to analyze pre-processed MSI data and other types of high dimensional data. However, for large scale MSI data analytics (e.g. $10^5 - 10^6$ spectra without prior peak picking), these methods will be slower, require extensive memory, and may become computationally intractable. We have applied these algorithms to the analysis of two large MSI datasets (3D MALDI MSI of Mouse Kidney and 3D MALDI MSI of human OSSC). Table 1 below summarizes the computational performance of these algorithms in comparison to our method. We have updated the main manuscript by including Table 1 and adding the following text:

“The computational performance of msiPL was compared to two state-of-the-art methods for non-linear dimensionality reduction, namely: 1- Uniform Manifold Approximation and Projection (umap) [McInnes et al. arXiv:1802.03426,2018], and 2- Hierarchical Stochastic Neighbor Embedding (HSNE) [Pezzotti et al. Computer Graphics Forum,2016] which is the scalable version of t-SNE. These methods have been effectively used to analyze different types of high-dimensional data [Becht et al. Nature biotechnology (2019); van Unen et al. Nature communications (2017); Korsunsky et al. Nature Methods (2019)], nevertheless they exhibit computational limitations for large scale MSI data analysis (e.g. MSI data with $10^5 - 10^6$ spectra without prior peak picking). Table 1 shows that the computational performance of msiPL outperforms both umap and HSNE for the analysis of large scale MSI data.”

Table 2. Non-linear algorithmic performance of MSI spectral data (time and memory comparison)

Dataset		umap	HSNE	msiPL
3D MALDI MSI of Mouse Kidney #Spec = 1,362,830 #m/z = 7671	Memory	Computationally Intractable	121.8 GB RAM	<6 GB RAM
	Time		43 min	Training: 8.6 min Testing: 10 sec/tissue
3D MALDI MSI of Human OSCC #Spec= 828,558 #m/z = 7665	Memory	155 GB RAM	90 GB RAM	<6 GB RAM
	Time	99.95 min	25 min	Training: 6.1 min Testing: 8 sec/tissue

We had originally compared the computational performance of spectral-wise peak picking using our method and a classical approach that utilizes the orthogonal matching pursuit algorithm (OMP) that is implemented in the Bruker software SCiLS Lab version 2020a. In the classical approach, the spectral-wise peak picking aims at increasing the identification sensitivity through analyzing a large number of spectra but at the cost of processing speed due to the OMP quadratic complexity. The results are summarized in the supplementary material (Table S7). Moreover, in our response to the first comment of Reviewer 1, we have performed a quantitative comparison of the peak picking performance between our method and a more rapid, but less sensitive classical approach for which peak picking was applied on the mean spectrum (see above Figure S11).

Table S7. Running Time for peak picking using commercial software compared to msiPL

Dataset	Running Time: Only Peak Picking Using SCiLS 2019c (Bruker, Germany)	The entire msiPL analysis (not just the peak learning step)
2D FT ICR MSI of human prostate cancer (12,716 spectra; 730,403 m/z bins) * *sampling every 20th spectrum for SCiLS analysis.	4 hours and 50 minutes	40 minutes
3D MALDI MSI of a PDX mouse brain model of glioblastoma (14,833 spectra; 661,402 m/z bins)+ +sampling every 5th spectrum for SCiLS analysis.	10 hours and 39 minutes	Training phase: 3.6 minutes Testing phase: 8 seconds. Note: 3D MSI dataset is analyzed using msiPL using training/testing strategy as explained in the manuscript.

2. The performance of the method was also tested using various biological samples collected from different organs; i) human prostate cancer tissue specimens, ii) a mouse brain model of glioblastoma, iii) human colorectal adenocarcinoma and iv) human oral squamous cell carcinoma. Molecular patterns associated with tumour regions were identified and ion features were tentatively 'identified' based on accurate mass. The authors tested the method on different tissue types but they did not use the method for defining different grades of e.g., human prostate cancer. Instead of providing data from many different cancer types it would have been much more interesting to test and define the method on different grades of the cancers. Please include an example(s).

Ans: We thank the reviewer for this insightful point. We are currently working on the development of methods for classification and predictive models. While the methods in development are beyond the scope of the presented study, we have here expanded our analysis to address the comment. We have applied the msiPL model trained with a MALDI FT-ICR MSI dataset acquired from a prostate cancer tissue section with Gleason score (3+4) (Figure 2) on a new unseen prostate tissue from a different patient with a distinct Gleason score (5+4). The testing dataset encompasses 13,471 spectra each of which has 61,343 m/z values. While the model training phase took 40 minutes, the overall testing phase took 56

seconds. The results are summarized in the Figure S13 below, which is now added to the supplementary material. The main manuscript has been updated with the following text:

“We sought to investigate the capabilities of a trained msiPL model to analyze unseen data of similar tumor type but from a different subject with different tumor grade. Here, the model trained with a MALDI FT-ICR MSI dataset from prostate cancerous tissue with a Gleason score (3+4) (Figure 2) was applied to the analysis of a distinct MALDI FT-ICR MSI dataset, see Supplementary Figure S13. The histopathological annotation of the test tissue section revealed two cancerous regions with distinct Gleason scores of (5+4) and (3+4) (Supplementary Figure S13-d). The test MSI dataset, constituting of 13,471 spectra each with 61,343 m/z values, was analyzed by the trained msiPL model in 56 seconds. The model was able to predict and reconstruct the original test data with an overall mean squared error of 2.273×10^{-5} , and the overlay of the average TIC normalized spectrum of both original and predicted data is shown in Supplementary Figure S13-a. The distribution of the GMM model selection criterion based on the Bayesian information criterion (BIC) and Kneedle algorithm revealed an optimal number of K-clusters (K=11) (Supplementary Figure S13-b), which was applied to a clustering analysis of the encoded features (Supplementary Figures S13-c and S13-e). Of interest, the model captured a spatial cluster associated with the histopathological annotation of Gleason score (5+4) (Supplementary Figure S13-f), and the Pearson correlation analysis revealed the highest correlated ion feature at m/z 786.5981 \pm 0.001 (Supplementary Figure S13-g). In accordance with a recent study that analyzed the same dataset, we noticed clustering distinction between two Gleason scores of (5+4) and (3+4) [Randall et al, Molecular Cancer Research, 2019]. While our results support the efficiency of msiPL for unsupervised mining of different MSI datasets and identification of spatial patterns of biological relevance, we envision future extensions of the msiPL model to enable classification and predictive tasks for tumor type and grade directly from the mass spectral data.”

Figure S13. Analysis of a test MALDI FT-ICR MSI dataset from prostate cancer tissue: **a**. Overlay of the mean spectrum of both TIC normalized original (green) and predicted (red) data with an overall mean squared error of 2.273×10^{-5} . **b**. Model selection of K -clusters to automatically cluster the encoded features shown in (c) using Bayesian information criterion (BIC) and the Kneedle algorithm. **d**. Histopathological annotation of the cancerous regions and associated Gleason score (GS). **e**. GMM-based clustering ($K=11$) of the encoded features (c) reveals a tumor cluster (f) that was found associated with a higher tumor grade region of GS (5+4). **g**. Spatial distribution of the highest correlated ion feature with the tumor cluster (f) was found at m/z 786.5981 ± 0.001 with a Pearson correlation coefficient of 0.746.

REVIEWER COMMENTS

Reviewer #1 (Remarks to the Author):

The authors have performed a revision and added new results, in particular by comparing their method against OMP (in terms of detected peaks) and against UMAP/HSNE in terms of memory and runtime.

However, there is still a big question whether their proposed method is better than the state of the art efficient methods for dimensionality reduction developed specifically for MSI as reviewed by Verbeeck et al 2019 Mass Spectrom Rev. These include: Memory-Efficient PCA, DWT, random projections, and compressed sensing (see section "B. Intermezzo: Dimensionality Reduction and Computational Resources" in Verbeeck et al 2019 Mass Spectrom Rev).

Below, I comment on how this revision addresses the four major issues raised by me in the original review.

First major issue: partially addressed

I welcome the performed comparison against a commercially available method (orthogonal matching pursuit (OMP) algorithm applied to a mean spectrum). The results confirm what I suspected: this 10-years old method is lightning fast and takes a few seconds only compared to 5+ minutes for the proposed method msiPL. Even with default parameters it finds 5% of unique peaks not found by msiPL. I'm pretty sure that by changing the OMP parameters from default the authors would find that it can detect almost all peaks detected by msiPL.

Moreover, the presentation of new results is biased and without error analysis: the authors show peaks detected by their algorithm only but do not show 5% of peaks detected by OMP.

Second major issue: partially addressed

Following my request to compare their method with the state of the art methods for dimensionality in particular from (Verbeeck et al 2019 Mass Spectrom Rev), the authors compared their method to UMAP and HSNE. This, however, does not address the raised issue, as the choice of UMAP nor HSNE is rather arbitrary and neither UMAP nor HSNE represent the most efficient state of the art methods in MSI. Instead, there is a number of memory/computing power-efficient methods specifically developed for MSI as reviewed by Verbeeck et al 2019 Mass Spectrom Rev. These include: Memory-Efficient PCA, DWT, random projections, and compressed sensing (see section "B. Intermezzo: Dimensionality Reduction and Computational Resources" in Verbeeck et al 2019 Mass Spectrom Rev).

I request once again to perform a comparison with the computationally efficient methods which correspond to the state of the art in MSI and are available e.g. the random projections code is available at <https://pubs.acs.org/doi/10.1021/ac400184g>.

Third major issue: not addressed

In their response, the authors avoid any comparison to other efficient methods for dimensionality reduction which were developed for MSI; let me cite again (Verbeeck et al 2019 Mass Spectrom Rev) and refer to examples I cited in the Second major issue.

Please do not get me wrong, I do not argue against using the mean squared error (MSE) as a metric for evaluating the quality. However, I request authors to provide evidence that their MSE is lower than can be achieved by the state of the art methods (Verbeeck et al 2019 Mass Spectrom Rev).

Fourth major issue: addressed

The code is indeed provided since August 2020. I'm sorry for claiming that it was not available, likely because during the earlier submission of this manuscript to Nature Methods it was not available and I didn't double check this particular point when submitting my review to Nature Communications.

Reviewer #2 (Remarks to the Author):

The authors addressed all issues appropriately.

REVIEWER COMMENTS AND RESPONSES MARCH 2021

Reviewer #1 (Remarks to the Author):

The authors have performed a revision and added new results, in particular by comparing their method against OMP (in terms of detected peaks) and against UMAP/HSNE in terms of memory and runtime.

However, there is still a big question whether their proposed method is better than the state of the art efficient methods for dimensionality reduction developed specifically for MSI as reviewed by Verbeeck et al 2019 Mass Spectrom Rev. These include: Memory-Efficient PCA, DWT, random projections, and compressed sensing (see section "B. Intermezzo: Dimensionality Reduction and Computational Resources" in Verbeeck et al 2019 Mass Spectrom Rev).

Below, I comment on how this revision addresses the four major issues raised by me in the original review.

To be more specific, we have already compared against OMP both **in terms of detected peaks and time**, as reported in our prior revision. Also, the dimensionality reduction approaches cited in the 2019 review are not state of the art. We used UMAP [published 2018; 2090 citations as of 03/02/21; first application of UMAP in *Nat Biotech* 12/03/19 **37**: 38-44 (2019) 969 citations as of 03/02/21] and HSNE [published in 2016; t-SNE 2008 18,148 citations as of 03/02/21; HSNE which is a scalable version for large datasets 2016; 163 citations] as state of the art while the reviewer is referring to older methods as state of the art. In fact, the first author of the cited review article, Verbeeck, just published a few days ago using UMAP as a dimensionality reduction approach (*Anal Bioanal Chem* 2021 Mar 1 *Spatially aware clustering of ion images in mass spectrometry imaging data using deep learning*, Zhang, Claesen, Moerman, Groseclose, Waelkens, De Moor, and Verbeeck). The reviewer refers to methods that are not state of the art, including DWT, which was published by Van de Plas in 2008 (<https://dl.acm.org/doi/10.1145/1363686.1363989>) and cited 11 times.

Here are the original references for algorithms that the Reviewer keeps referring to as "state of the art" based on their cited review article in the journal *Mass Spectrometry Reviews*:

- 1- PCA [1987] by Wold, Svante, Kim Esbensen, and Paul Geladi. "Principal component analysis." *Chemometrics and intelligent laboratory systems* 2.1-3 (1987): 37-52.
- 2- Random Projections [2001]: Bingham, Ella, and Heikki Mannila. "Random projection in dimensionality reduction: applications to image and text data." *Proceedings of the seventh ACM SIGKDD international conference on Knowledge discovery and data mining*. 2001.
- 3- DWT[1992] Shensa, Mark J. "The discrete wavelet transform: wedding the a trous and Mallat algorithms." *IEEE Transactions on signal processing* 40.10 (1992): 2464-2482.

First major issue: partially addressed

I welcome the performed comparison against a commercially available method (orthogonal matching pursuit (OMP) algorithm applied to a mean spectrum). The results confirm what I suspected: this 10-years old method is lightning fast and takes a few seconds only compared to 5+ minutes for the proposed method msiPL. Even with default parameters it finds 5% of unique peaks not found by msiPL. I'm pretty sure that by changing the OMP parameters from default the authors would find that it can detect almost all peaks detected by msiPL.

Moreover, the presentation of new results is biased and without error analysis: the authors show peaks detected by their algorithm only but do not show 5% of peaks detected by OMP.

First, we assume that the reviewer meant 9% above, and not 5%. Also, we had already specifically expanded in the revised manuscript on the balance between analysis time and peak identification. Moreover, we had also provided the extensive list of peaks found by OMP and msiPL, referred again here "The peaks identified by both msiPL and the OMP

are highlighted in Figure S11.a, and listed in the Supplementary File “Peaks_msiPL_OMP(MeanSpec).xlsx””. Results from both were also highlighted in Figure S.11.

Again, from page 253 of Verbeeck, Caprioli, van de Plas: “Peak picking is, however, a rather drastic form of feature selection that discards a large amount of information from the original data (e.g., peak shape), while also holding the risk of discarding peaks that go unrecognized by the peak-picking algorithm. This makes the quality of the subsequent analysis dependent on the quality of the preceding peak-picking or feature selection method, which may not always be desirable (Palmer, Bunch, & Styles, 2013, 2015).” One significant advance with msiPL is to avoid the optimization of preprocessing parameters which are known to affect downstream analyses, as stated in your own cited review above (Verbeeck et al 2019 Mass Spectrom Rev). We also highlighted that parameters could be modified and produce different results, but again, an advantage of msiPL is to avoid the subjective step while producing reliable results in a timely manner. We have now edited Figure S11 by adding a pie chart comparison that showed that msiPL covered 99% of peaks identified by a more accurate approach of applying OMP algorithm on thousands of spectra which comes at the cost of slow processing as shown in Table S7 (which was already presented in the prior revision).

Table S7. Running Time for peak picking using commercial software compared to msiPL

Dataset	Running time for only peak picking step using OMP algorithm implemented in SCiLS 2019c (Bruker, Germany)	The entire msiPL analysis (not just the peak learning step)
2D FT ICR MSI of human prostate cancer (12,716 spectra; 730,403 m/z bins) * *sampling every 20th spectrum for OMP analysis.	4 hours and 50 minutes	40 minutes
3D MALDI MSI of a PDX mouse brain model of glioblastoma (14,833 spectra; 661,402 m/z bins)+ +sampling every 5th spectrum for OMP analysis.	10 hours and 39 minutes	Training phase: 3.6 minutes Testing phase: 8 seconds. Note: 3D MSI dataset is analyzed using msiPL using training/testing strategy as explained in the manuscript.

FigureS11. Comparison of peak picking analysis on the 3D MALDI FT-ICR MSI PDX GBM dataset: (a-b) using msiPL on full spectral data and the orthogonal matching pursuit (OMP) algorithm on the mean spectrum (implemented in the commercial software of SCI LS Lab version 2020a (Bruker, Bremen, Germany)). (c-d) visualization of some of the peaks with biological relevance identified only by the manifold learning approach of msiPL but not by a classical approach of applying the OMP algorithm on the mean spectrum. (e) msiPL covered 99% of peaks identified by a more powerful approach of applying the OMP method on a wider range of spectra (thousands) but at cost of slow processing (Table S7).

Following my request to compare their method with the state of the art methods for dimensionality in particular from (Verbeeck et al 2019 Mass Spectrom Rev), the authors compared their method to UMAP and HSNE. This, however, does not address the raised issue, as the choice of UMAP nor HSNE is rather arbitrary and neither UMAP nor HSNE represent the most efficient state of the art methods in MSI. Instead, there is a number of memory/computing power-efficient methods specifically developed for MSI as reviewed by Verbeeck et al 2019 Mass Spectrom Rev. These include: Memory-Efficient PCA, DWT, random projections, and compressed sensing (see section "B. Intermezzo: Dimensionality Reduction and Computational Resources" in Verbeeck et al 2019 Mass Spectrom Rev).

I request once again to perform a comparison with the computationally efficient methods which correspond to the state of the art in MSI and are available e.g. the random projections code is available at <https://pubs.acs.org/doi/10.1021/ac400184g>

The random projections code cited above was published in 2013 and received 16 citations. This is not state of the art. Verbeeck, first author of the reviewer's referred review article, published a few days ago, March 1st 2021 using UMAP as a state of the art dimensionality reduction tool. PCA and DWT (published in 2008 by Van de plas and cited 11 times) are not state of the art. We have compared/benchmarked msiPL to state of the art approaches.

Please also note the comment from Race *et al.* for the Reviewer's referred method "Memory Efficient Principal Component Analysis for the Dimensionality Reduction of Large Mass Spectrometry Imaging Data Sets", *Anal Chem* 2013, which is as follows: "Note that the full covariance matrix must be constructed and very high-dimensional data sets may still prove to be intractable. For this reason, we employ peak detection methods in order to reduce the dimensionality of the data."

Third major issue: not addressed

In their response, the authors avoid any comparison to other efficient methods for dimensionality reduction which were developed for MSI; let me cite again (Verbeeck et al 2019 Mass Spectrom Rev) and refer to examples I cited in the Second major issue. Please do not get me wrong, I do not argue against using the mean squared error (MSE) as a metric for evaluating the quality. However, I request authors to provide evidence that their MSE is lower than can be achieved by the state of the art methods (Verbeeck et al 2019 Mass Spectrom Rev).

Again, our study does not present a new method for dimensionality reduction, but an integrated method for pre-processing and dimensionality reduction bypassing preprocessing parameters optimization. The publication referred to by the reviewer acknowledges the limitations imposed by preprocessing on downstream dimensionality reduction as cited above from page 253 of Verbeeck et al 2019 Mass Spectrom Rev. Our study proposes an integrated workflow that avoids the need for parameter optimization in preprocessing. This is an issue acknowledged by many in the field including a recent *Nature Communications* report (*Nat Commun.* 2020 Nov 5;11(1):5595. doi: 10.1038/s41467-020-19354-z. PMID: 33154370) and other fields such as image analysis (Isensee, F., Jaeger, P.F., Kohl, S.A.A. *et al.* nnU-Net: a self-configuring method for deep learning-based biomedical image segmentation. *Nat Methods* **18**, 203–211 (2021). <https://doi.org/10.1038/s41592-020-01008-z>). Nevertheless, please see the requested comparison in the following table, included in the manuscript as Suppl. Table S8.

This text is now added to the main manuscript: "The reconstruction error of msiPL surpassed other methods that were previously applied on MSI data¹⁸ such as PCA, memory efficient PCA, and Discrete wavelet transform (DWT) followed by PCA. see Supplementary Table S8."

Table S8. Comparison of mean squared error (MSE) of MSI data reconstruction using different methods

	PCA	Memory Efficient PCA	DWT + PCA	msiPL
FT ICR MSI Prostate Dataset	1.886×10^{-2}	2.583×10^{-2}	1.81×10^{-2}	2.42×10^{-5}
FT ICR MSI GBM Dataset	8.405×10^{-3}	7.018×10^{-3}	8.6×10^{-3}	4.5×10^{-4}

Fourth major issue: addressed

The code is indeed provided since August 2020. I'm sorry for claiming that it was not available, likely because during the earlier submission of this manuscript to Nature Methods it was not available and I didn't double check this particular point when submitting my review to Nature Communications.

Our code was provided since our initial submission to *Nature Methods* in 15th November 2019, which was acknowledged by the other reviewer in the reviews we had received from *Nature Methods* in January 2020 (Documented in the attached *Nature Methods* letters).

Reviewer #2 (Remarks to the Author):

The authors addressed all issues appropriately.

Thank you.

REVIEWER COMMENTS

Reviewer #3 (Remarks to the Author):

I am sharing what I wrote to the editor. I was not involved with the reviews at any stage so it is an outsider opinion. I am truly appreciative that the editor is seeking another opinion with respect to a disagreement between the authors and reviewers. I don't envy the editor in this case as it is hard. For one, you have work that is done at a very high level and then on the other hand you have the writing and accurate review that are in disagreement. What is rather unfortunate is that this is largely due to the language the authors choose to use that invites the requirement of the comparisons requested and in combination with an apparent lack of awareness, appreciation, or acknowledgment of what is done outside of their lab.

I am only evaluating the discussion in the context of the paper and not the science of the paper itself. The reviewer is quite fair based on the paper as written and makes very good points and seems genuinely interested in ensuring the statements made in the paper are accurate. I also know as an author of a lot of methods papers that comparisons to everything under the sun is not realistic and even impossible.

It is clear that the authors are super excited -and justifiably so- about these developments for their lab and the images and figures presented are spectacular but missing a new biological discovery that this has enabled. It does not mean that the methods paper does not stand on its own but the challenge is that the authors make unnecessary claims and then do not back those claims up with data. Basically, the way they wrote their paperback themselves in a corner to prove their statements-which were no doubt emerged about the enthusiasm of ease of use and capabilities when compared to what they used in their own lab previously but as written they invite comparisons to existing tools, especially since they seem to not have been aware of the availability and not valuing other scientific contributions.

Just in the abstract as well as their paper. They use a lot of largely unsupported claims. Just in the abstract "robust", "efficiently", "few seconds", "memory-efficient implementation", "significantly less memory", all require comparisons to existing tools as the reviewer appropriately did. These are also terms not needed for this work. One million high dimensional data analysis has been done for more than a decade. It states less memory but then one needs to compare to all methods that claim the same for comparison. Compare this to -for example- the abstract in a related field. <https://www.nature.com/articles/s41587-021-00860-4>, there are also key and related claims but that are backed up with clear numbers, this is not the case in the current paper. They showcased it was able to report on things that was not previously possible. The current paper does not seem to do this rather it focuses on an autoencoder solution but unclear what they were able to do with it that previously could not.

The rationale by the authors not comparing to other methods based on their interpretation as being state-of-the-art or not based on citations is flawed at best, especially since this is a relatively small field. The scale of analysis of IMS data has been done for quite some time-perhaps as long as 15 years. I have written papers that dealt with this volume of MSI data myself-if not ten times that. To me it reads that the authors want to argue their way-out vs addressing topics head-on yet keeping their overstated claims. This work reminds me of what is possible in SCILS, a 2D and 3D imaging MS visualization program commercially available from Bruker, or even Raf van der Plas work in the past <https://www.ncbi.nlm.nih.gov/pmc/articles/PMC4382398/> that although did not use the term autoencoder it is a training and then high-resolution prediction. The term autoencoder was not common in 2015 but it is a training and reporting type scenario. Perhaps most relevant is a computer science conference proceedings (which are the some of the most important papers in such a field and will have 5-6 reviewers before it is accepted and only has 5% acceptance rates-something that life scientists often do not recognize) that do autoencoder based processing in imaging mass spec <https://ieeexplore.ieee.org/document/7849863>, <https://academic.oup.com/bioinformatics/article/34/7/1215/4604594> <https://pubs.rsc.org/en/content/articlehtml/2017/sc/c6sc03738k> <https://pubs.acs.org/doi/abs/10.1021/acs.jproteome.7b00725> reviewed in

https://www.annualreviews.org/doi/abs/10.1146/annurev-biodatasci-011420-031537?casa_token=pxiaY7gsp84AAAAA%3Atn3rufmpRTbcQiub4UPG68TSWt6ZoezfAskvaEv_TQuYMiVU_uCZnLRI7hkokrCB1NEEFPAi_oPyg

This is a paper that is a cog in the gearbox to make the engine better. I do not think it is necessary to make all the comparisons with existing literature tools. This is somewhat of an unreasonable request as it is not known what parameters to compare and if you compare one metric lest say speed-this does not inform on quality as there is no ground truth of what images to expect. The underpinnings of the work appear well-executed and appropriate for Nat communications if they can find a way to remove all those unnecessary claims and acknowledge what the literature has done. The more such tools are developed the better it is for the community. I wonder if their own laboratory even uses these tools. I.e. do any recent papers from their lab show the utility-this would make this work even more convincing. My advice is to recommend them to cut the abstract to 150 words, remove qualifiers that require comparisons from abstract and text. The abstract should highlight that “despite many solutions being developed (this means they can also accurately review the lit in the intro), that it is still hard to deal with the large volumes of data and information generated in imaging, especially 3D data sets”. And that they, therefore, assessed if “a probabilistic generative model based on a fully-connected variational autoencoder could be used for unsupervised analysis and peak learning of MSI data” to help uncover hidden and create sharper images. They should provide a rationale for why they believed this approach would be useful in the analysis. And then what their key observations are – list capabilities of the tool with hard data. And then closing statement. Currently, the paper almost reads we want to use machine learning without a good rationale, especially in the context of the literature, this should be avoided.

I do not understand the author's approach to the reviews in this case. The hard part is that reviewer one based on the history of how the authors responded – completely dismissive and not take their advice serious – are less likely to support a publication even if reformatted. This is making this incredibly hard for the Editor to make a decision on something that really falls on the shoulders of the authors. Not even a simple thank you for their precious time reviewing papers, an acknowledgment that they found good papers that probably should be cited etc are simple things that can help a lot. It is OK to disagree with reviewers but then acknowledge when they are right as well. For example. “The results confirm what I suspected: this 10-years old method is lightning fast and takes a few seconds only compared to 5+ minutes for the proposed method msiPL. Even with default parameters, it finds 5% of unique peaks not found by msiPL.” This is a very telling passage and how the authors responded where the key first point is to argue it was 9% not 5% as opposed to saying that it was good intuition by the reviewer and agree that speed is not the key factor and was overstated in the abstract. The key advantages are XXXX,YYYY, ZZZZ. And this is how we adjusted the text accordingly and following citations have been added. So instead they argued against what they have now demonstrated to be a scientifically flawed statement in their paper to still be included instead of addressing it.

Finally, with Metabolomics WB, Figshare, Metaspace

“The 2D MALDI FT ICR MSI data of a human prostate tissue, and the 3D MALDI FT ICR MSI data of a PDX mouse brain model of glioblastoma are available from the corresponding author on reasonable request” is unacceptable and data should be made public in the same way they benefited from the public data by Oetjes et al. I would personally also not accept a paper without public access to the data. imzML is acceptable as well.

Response To Reviewer #3 (Remarks to the Author):

We would like to sincerely thank the Reviewer for the evident effort that was invested in providing an outside opinion to the review of our manuscript and we are extremely grateful for the constructive comments brought by the Reviewer, though embarrassed by the tone of our prior response. Thank you for the transparency in sharing comments to the Editor. We here focus our response to address the two specific requests from the Reviewer and have removed the comments to the Editor.

My advice is to recommend them to cut the abstract to 150 words, remove qualifiers that require comparisons from abstract and text. The abstract should highlight that “despite many solutions being developed (this means they can also accurately review the lit in the intro), that it is still hard to deal with the large volumes of data and information generated in imaging, especially 3D data sets”. And that they, therefore, assessed if “a probabilistic generative model based on a fully-connected variational autoencoder could be used for unsupervised analysis and peak learning of MSI data” to help uncover hidden and create sharper images. They should provide a **rationale for why they believed this approach would be useful in the analysis**. And then what their key observations are – **list capabilities of the tool with hard data. And then closing statement**. Currently, the paper almost reads we want to use machine learning without a good rationale, especially in the context of the literature, this should be avoided.

Thank you for such constructive recommendation. We have restructured the abstract accordingly and reduced to 176 words. We have also removed qualifiers that require comparison throughout the manuscript (we highlighted all the sentences where the changes were made). We have also updated references in the Introduction according to the Reviewer’s recommendation.

Reduced abstract:

“Mass spectrometry imaging (MSI) is an emerging technology that holds potential for improving, biomarker discovery, metabolomics research, pharmaceutical applications and clinical diagnosis. Despite many solutions being developed, the large data size and high dimensional nature of MSI, especially 3D datasets, still pose computational and memory complexities that hinder accurate identification of biologically relevant molecular patterns. Moreover, the subjectivity in the selection of parameters for conventional pre-processing approaches can lead to bias. Therefore, we assessed if a probabilistic generative model based on a fully connected variational autoencoder could be used for unsupervised analysis and peak learning of MSI data to uncover hidden structures. The resulting msiPL method could learn and visualize the underlying non-linear spectral manifold, reveal biologically relevant clusters of tissue anatomy in a mouse kidney and tumor heterogeneity in human prostatectomy tissue, colorectal carcinoma, and glioblastoma mouse model, with identification of underlying m/z peaks. The method was applied for the analysis of MSI datasets ranging from 3.3 to 78.9 GB, without prior pre-processing and peak picking, and acquired using different mass spectrometers at different centers.”

Finally, “The 2D MALDI FT ICR MSI data of a human prostate tissue, and the 3D MALDI FT ICR MSI data of a PDX mouse brain model of glioblastoma are available from the corresponding author on reasonable request” is unacceptable and data should be made public in the same way they benefited from the public data by Oetjes et al. I would personally also not accept a paper without public access to the data. imzML is acceptable as well.

We are delighted to report that we were able to obtain clearance from our institutional IRB to publicly release the human prostate tissue dataset. We have now deposited both datasets on the UCSD Metabolomics Workbench Data Repository, namely the human prostate and 3D FT ICR PDX mouse brain model of glioblastoma MSI data in Hierarchical Data Format (hdf5). The following statement is now added to the main manuscript and will be updated with a direct link once assigned:

“This data is available at the NIH Common Fund's National Metabolomics Data Repository (NMDR) website, the Metabolomics Workbench, <https://www.metabolomicsworkbench.org> where it has been assigned Project ID (2703). The data can be accessed directly via this link https://www.metabolomicsworkbench.org/data/MWTABMetadata4.php?Mode=Study&DataMode=AllData&StudyType=MS&F=wabdelmoula_20210620_173225_mwtab_analysis_1.txt